# Linear Convergent Decentralized Optimization with Compression

**Xiaorui Liu**[1], **Yao Li**[2,3], **Rongrong Wang**[3,2], **Jiliang Tang**[1] **& Ming Yan**[3,2]

[1] Department of Computer Science and Engineering
[2] Department of Mathematics
[3] Department of Computational Mathematics, Science and Engineering
Michigan State University, East Lansing, MI 48823, USA
`{xiaorui,liyao6,wongron6,tangjili,myan}@msu.edu`

## Abstract

Communication compression has become a key strategy to speed up distributed optimization. However, existing decentralized algorithms with compression mainly focus on compressing DGD-type algorithms. They are unsatisfactory in terms of convergence rate, stability, and the capability to handle heterogeneous data. Motivated by primal-dual algorithms, this paper proposes the first LinEAr convergent Decentralized algorithm with compression, LEAD. Our theory describes the coupled dynamics of the inexact primal and dual update as well as compression error, and we provide the first consensus error bound in such settings without assuming bounded gradients. Experiments on convex problems validate our theoretical analysis, and empirical study on deep neural nets shows that LEAD is applicable to non-convex problems.

## 1 Introduction

Distributed optimization solves the following optimization problem

$$\boldsymbol{x}^* := \arg\min_{\boldsymbol{x} \in \mathbb{R}^d} \left[ f(\boldsymbol{x}) := \frac{1}{n} \sum_{i=1}^{n} f_i(\boldsymbol{x}) \right] \tag{1}$$

with $n$ computing agents and a communication network. Each $f_i(\boldsymbol{x}) : \mathbb{R}^d \to \mathbb{R}$ is a local objective function of agent $i$ and typically defined on the data $\mathcal{D}_i$ settled at that agent. The data distributions $\{\mathcal{D}_i\}$ can be heterogeneous depending on the applications such as in federated learning. The variable $\boldsymbol{x} \in \mathbb{R}^d$ often represents model parameters in machine learning. A distributed optimization algorithm seeks an optimal solution that minimizes the overall objective function $f(\boldsymbol{x})$ collectively. According to the communication topology, existing algorithms can be conceptually categorized into centralized and decentralized ones. Specifically, centralized algorithms require global communication between agents (through central agents or parameter servers). While decentralized algorithms only require local communication between connected agents and are more widely applicable than centralized ones. In both paradigms, the computation can be relatively fast with powerful computing devices; efficient communication is the key to improve algorithm efficiency and system scalability, especially when the network bandwidth is limited.

In recent years, various communication compression techniques, such as quantization and sparsification, have been developed to reduce communication costs. Notably, extensive studies (Seide et al., 2014; Alistarh et al., 2017; Bernstein et al., 2018; Stich et al., 2018; Karimireddy et al., 2019; Mishchenko et al., 2019; Tang et al., 2019b; Liu et al., 2020) have utilized gradient compression to significantly boost communication efficiency for centralized optimization. They enable efficient large-scale optimization while maintaining comparable convergence rates and practical performance with their non-compressed counterparts. This great success has suggested the potential and significance of communication compression in decentralized algorithms.

While extensive attention has been paid to centralized optimization, communication compression is relatively less studied in decentralized algorithms because the algorithm design and analysis are

more challenging in order to cover general communication topologies. There are recent efforts trying to push this research direction. For instance, DCD-SGD and ECD-SGD (Tang et al., 2018a) introduce difference compression and extrapolation compression to reduce model compression error. (Reisizadeh et al., 2019a;b) introduce QDGD and QuanTimed-DSGD to achieve exact convergence with small stepsize. DeepSqueeze (Tang et al., 2019a) directly compresses the local model and compensates the compression error in the next iteration. CHOCO-SGD (Koloskova et al., 2019; 2020) presents a novel quantized gossip algorithm that reduces compression error by difference compression and preserves the model average. Nevertheless, most existing works focus on the compression of primal-only algorithms, i.e., reduce to DGD (Nedic & Ozdaglar, 2009; Yuan et al., 2016) or P-DSGD (Lian et al., 2017). They are unsatisfying in terms of convergence rate, stability, and the capability to handle heterogeneous data. Part of the reason is that they inherit the drawback of DGD-type algorithms, whose convergence rate is slow in heterogeneous data scenarios where the data distributions are significantly different from agent to agent.

In the literature of decentralized optimization, it has been proved that primal-dual algorithms can achieve faster converge rates and better support heterogeneous data (Ling et al., 2015; Shi et al., 2015; Li et al., 2019; Yuan et al., 2020). However, it is unknown whether communication compression is feasible for primal-dual algorithms and how fast the convergence can be with compression. In this paper, we attempt to bridge this gap by investigating the communication compression for primal-dual decentralized algorithms. Our major contributions can be summarized as:

- We delineate two key challenges in the algorithm design for communication compression in decentralized optimization, i.e., data heterogeneity and compression error, and motivated by primal-dual algorithms, we propose a novel decentralized algorithm with compression, LEAD.

- We prove that for LEAD, a constant stepsize in the range $(0, 2/(\mu + L)]$ is sufficient to ensure linear convergence for strongly convex and smooth objective functions. To the best of our knowledge, LEAD is the first linear convergent decentralized algorithm with compression. Moreover, LEAD provably works with unbiased compression of arbitrary precision.

- We further prove that if the stochastic gradient is used, LEAD converges linearly to the $O(\sigma^2)$ neighborhood of the optimum with constant stepsize. LEAD is also able to achieve exact convergence to the optimum with diminishing stepsize.

- Extensive experiments on convex problems validate our theoretical analyses, and the empirical study on training deep neural nets shows that LEAD is applicable for nonconvex problems. LEAD achieves state-of-art computation and communication efficiency in all experiments and significantly outperforms the baselines on heterogeneous data. Moreover, LEAD is robust to parameter settings and needs minor effort for parameter tuning.

## 2 RELATED WORKS

Decentralized optimization can be traced back to the work by Tsitsiklis et al. (1986). DGD (Nedic & Ozdaglar, 2009) is the most classical decentralized algorithm. It is intuitive and simple but converges slowly due to the diminishing stepsize that is needed to obtain the optimal solution (Yuan et al., 2016). Its stochastic version D-PSGD (Lian et al., 2017) has been shown effective for training nonconvex deep learning models. Algorithms based on primal-dual formulations or gradient tracking are proposed to eliminate the convergence bias in DGD-type algorithms and improve the convergence rate, such as D-ADMM (Mota et al., 2013), DLM (Ling et al., 2015), EXTRA (Shi et al., 2015), NIDS (Li et al., 2019), $D^2$ (Tang et al., 2018b), Exact Diffusion (Yuan et al., 2018), OPTRA(Xu et al., 2020), DIGing (Nedic et al., 2017), GSGT (Pu & Nedić, 2020), etc.

Recently, communication compression is applied to decentralized settings by Tang et al. (2018a). It proposes two algorithms, i.e., DCD-SGD and ECD-SGD, which require compression of high accuracy and are not stable with aggressive compression. Reisizadeh et al. (2019a;b) introduce QDGD and QuanTimed-DSGD to achieve exact convergence with small stepsize and the convergence is slow. DeepSqueeze (Tang et al., 2019a) compensates the compression error to the compression in the next iteration. Motivated by the quantized average consensus algorithms, such as (Carli et al., 2010), the quantized gossip algorithm CHOCO-Gossip (Koloskova et al., 2019) converges linearly to the consensual solution. Combining CHOCO-Gossip and D-PSGD leads to a decentralized algorithm with compression, CHOCO-SGD, which converges sublinearly under the strong convexity and

gradient boundedness assumptions. Its nonconvex variant is further analyzed in (Koloskova et al., 2020). A new compression scheme using the modulo operation is introduced in (Lu & De Sa, 2020) for decentralized optimization. A general algorithmic framework aiming to maintain the linear convergence of distributed optimization under compressed communication is considered in (Magnússon et al., 2020). It requires a contractive property that is not satisfied by many decentralized algorithms including the algorithm in this paper.

## 3 ALGORITHM

We first introduce notations and definitions used in this work. We use bold upper-case letters such as $\mathbf{X}$ to define matrices and bold lower-case letters such as $\boldsymbol{x}$ to define vectors. Let $\mathbf{1}$ and $\mathbf{0}$ be vectors with all ones and zeros, respectively. Their dimensions will be provided when necessary. Given two matrices $\mathbf{X}$, $\mathbf{Y} \in \mathbb{R}^{n \times d}$, we define their inner product as $\langle \mathbf{X}, \mathbf{Y} \rangle = \mathrm{tr}(\mathbf{X}^\top \mathbf{Y})$ and the norm as $\|\mathbf{X}\| = \sqrt{\langle \mathbf{X}, \mathbf{X} \rangle}$. We further define $\langle \mathbf{X}, \mathbf{Y} \rangle_\mathbf{P} = \mathrm{tr}(\mathbf{X}^\top \mathbf{P} \mathbf{Y})$ and $\|\mathbf{X}\|_\mathbf{P} = \sqrt{\langle \mathbf{X}, \mathbf{X} \rangle_\mathbf{P}}$ for any given symmetric positive semidefinite matrix $\mathbf{P} \in \mathbb{R}^{n \times n}$. For simplicity, we will majorly use the matrix notation in this work. For instance, each agent $i$ holds an individual estimate $\boldsymbol{x}_i \in \mathbb{R}^d$ of the global variable $\boldsymbol{x} \in \mathbb{R}^d$. Let $\mathbf{X}^k$ and $\nabla \mathbf{F}(\mathbf{X}^k)$ be the collections of $\{\boldsymbol{x}_i^k\}_{i=1}^n$ and $\{\nabla f_i(\boldsymbol{x}_i^k)\}_{i=1}^n$ which are defined below:

$$\mathbf{X}^k = \left[\boldsymbol{x}_1^k, \ldots, \boldsymbol{x}_n^k\right]^\top \in \mathbb{R}^{n \times d}, \quad \nabla \mathbf{F}(\mathbf{X}^k) = \left[\nabla f_1(\boldsymbol{x}_1^k), \ldots, \nabla f_n(\boldsymbol{x}_n^k)\right]^\top \in \mathbb{R}^{n \times d}. \quad (2)$$

We use $\nabla \mathbf{F}(\mathbf{X}^k; \xi^k)$ to denote the stochastic approximation of $\nabla \mathbf{F}(\mathbf{X}^k)$. With these notations, the update $\mathbf{X}^{k+1} = \mathbf{X}^k - \eta \nabla \mathbf{F}(\mathbf{X}^k; \xi^k)$ means that $\boldsymbol{x}_i^{k+1} = \boldsymbol{x}_i^k - \eta \nabla f_i(\boldsymbol{x}_i^k; \xi_i^k)$ for all $i$. In this paper, we need the average of all rows in $\mathbf{X}^k$ and $\nabla \mathbf{F}(\mathbf{X}^k)$, so we define $\overline{\mathbf{X}}^k = (\mathbf{1}^\top \mathbf{X}^k)/n$ and $\overline{\nabla} \mathbf{F}(\mathbf{X}^k) = (\mathbf{1}^\top \nabla \mathbf{F}(\mathbf{X}^k))/n$. They are row vectors, and we will take a transpose if we need a column vector. The pseudoinverse of a matrix $\mathbf{M}$ is denoted as $\mathbf{M}^\dagger$. The largest, $i$th-largest, and smallest nonzero eigenvalues of a symmetric matrix $\mathbf{M}$ are $\lambda_{\max}(\mathbf{M})$, $\lambda_i(\mathbf{M})$, and $\lambda_{\min}(\mathbf{M})$.

**Assumption 1** (Mixing matrix). *The connected network $\mathcal{G} = \{\mathcal{V}, \mathcal{E}\}$ consists of a node set $\mathcal{V} = \{1, 2, \ldots, n\}$ and an undirected edge set $\mathcal{E}$. The primitive symmetric doubly-stochastic matrix $\mathbf{W} = [w_{ij}] \in \mathbb{R}^{n \times n}$ encodes the network structure such that $w_{ij} = 0$ if nodes $i$ and $j$ are not connected and cannot exchange information.*

Assumption 1 implies that $-1 < \lambda_n(\mathbf{W}) \le \lambda_{n-1}(\mathbf{W}) \le \cdots \lambda_2(\mathbf{W}) < \lambda_1(\mathbf{W}) = 1$ and $\mathbf{W}\mathbf{1} = \mathbf{1}$ (Xiao & Boyd, 2004; Shi et al., 2015). The matrix multiplication $\mathbf{X}^{k+1} = \mathbf{W}\mathbf{X}^k$ describes that agent $i$ takes a weighted sum from its neighbors and itself, i.e., $\boldsymbol{x}_i^{k+1} = \sum_{j \in \mathcal{N}_i \cup \{i\}} w_{ij} \boldsymbol{x}_j^k$, where $\mathcal{N}_i$ denotes the neighbors of agent $i$.

### 3.1 THE PROPOSED ALGORITHM

The proposed algorithm LEAD to solve problem (1) is showed in Alg. 1 with matrix notations for conciseness. We will refer to the line number in the analysis. A complete algorithm description from the agent's perspective can be found in Appendix A. The motivation behind Alg. 1 is to achieve two goals: (a) consensus ($\boldsymbol{x}_i^k - (\overline{\mathbf{X}}^k)^\top \to \mathbf{0}$) and (b) convergence ($(\overline{\mathbf{X}}^k)^\top \to \boldsymbol{x}^*$). We first discuss how goal (a) leads to goal (b) and then explain how LEAD fulfills goal (a).

In essence, LEAD runs the approximate SGD globally and reduces to the exact SGD under consensus. One key property for LEAD is $\mathbf{1}_{n \times 1}^\top \mathbf{D}^k = \mathbf{0}$, regardless of the compression error in $\hat{\mathbf{Y}}^k$. It holds because that for the initialization, we require $\mathbf{D}^1 = (\mathbf{I} - \mathbf{W})\mathbf{Z}$ for some $\mathbf{Z} \in \mathbb{R}^{n \times d}$, e.g., $\mathbf{D}^1 = \mathbf{0}^{n \times d}$, and that the update of $\mathbf{D}^k$ ensures $\mathbf{D}^k \in \mathbf{Range}(\mathbf{I} - \mathbf{W})$ for all $k$ and $\mathbf{1}_{n \times 1}^\top (\mathbf{I} - \mathbf{W}) = \mathbf{0}$ as we will explain later. Therefore, multiplying $(1/n)\mathbf{1}_{n \times 1}^\top$ on both sides of Line 7 leads to a global average view of Alg. 1:

$$\overline{\mathbf{X}}^{k+1} = \overline{\mathbf{X}}^k - \eta \overline{\nabla} \mathbf{F}(\mathbf{X}^k; \xi^k), \quad (3)$$

which doesn't contain the compression error. Note that this is an approximate SGD step because, as shown in (2), the gradient $\nabla \mathbf{F}(\mathbf{X}^k; \xi^k)$ is not evaluated on a global synchronized model $\overline{\mathbf{X}}^k$. However, if the solution converges to the consensus solution, i.e., $\boldsymbol{x}_i^k - (\overline{\mathbf{X}}^k)^\top \to \mathbf{0}$, then $\mathbb{E}_{\xi^k}[\overline{\nabla} \mathbf{F}(\mathbf{X}^k; \xi^k) - \nabla f(\overline{\mathbf{X}}^k; \xi^k)] \to \mathbf{0}$ and (3) gradually reduces to exact SGD.

---

**Algorithm 1** LEAD

---

**Input:** Stepsize $\eta$, parameter $(\alpha, \gamma)$, $\mathbf{X}^0$, $\mathbf{H}^1$, $\mathbf{D}^1 = (\mathbf{I} - \mathbf{W})\mathbf{Z}$ for any $\mathbf{Z}$

**Output:** $\mathbf{X}^K$ or $1/n \sum_{i=1}^n \mathbf{X}_i^K$

1: $\mathbf{H}_w^1 = \mathbf{W}\mathbf{H}^1$                                 9: **procedure** COMM$(\mathbf{Y}, \mathbf{H}, \mathbf{H}_w)$

2: $\mathbf{X}^1 = \mathbf{X}^0 - \eta\nabla\mathbf{F}(\mathbf{X}^0; \xi^0)$                10:     $\mathbf{Q} = $ COMPRESS$(\mathbf{Y} - \mathbf{H})$

3: **for** $k = 1, 2, \cdots, K - 1$ **do**              11:     $\hat{\mathbf{Y}} = \mathbf{H} + \mathbf{Q}$

4:     $\mathbf{Y}^k = \mathbf{X}^k - \eta\nabla\mathbf{F}(\mathbf{X}^k; \xi^k) - \eta\mathbf{D}^k$      12:     $\hat{\mathbf{Y}}_w = \mathbf{H}_w + \mathbf{W}\mathbf{Q}$

5:     $\hat{\mathbf{Y}}^k, \hat{\mathbf{Y}}_w^k, \mathbf{H}^{k+1}, \mathbf{H}_w^{k+1} = $ COMM$(\mathbf{Y}^k, \mathbf{H}^k, \mathbf{H}_w^k)$     13:     $\mathbf{H} = (1 - \alpha)\mathbf{H} + \alpha\hat{\mathbf{Y}}$

6:     $\mathbf{D}^{k+1} = \mathbf{D}^k + \frac{\gamma}{2\eta}(\hat{\mathbf{Y}}^k - \hat{\mathbf{Y}}_w^k)$        14:     $\mathbf{H}_w = (1 - \alpha)\mathbf{H}_w + \alpha\hat{\mathbf{Y}}_w$

7:     $\mathbf{X}^{k+1} = \mathbf{X}^k - \eta\nabla\mathbf{F}(\mathbf{X}^k; \xi^k) - \eta\mathbf{D}^{k+1}$     15:     **Return:** $\hat{\mathbf{Y}}, \hat{\mathbf{Y}}_w, \mathbf{H}, \mathbf{H}_w$

8: **end for**                                       16: **end procedure**

---

With the establishment of how consensus leads to convergence, the obstacle becomes how to achieve consensus under local communication and compression challenges. It requires addressing two issues, i.e., data heterogeneity and compression error. To deal with these issues, existing algorithms, such as DCD-SGD, ECD-SGD, QDGD, DeepSqueeze, Moniqua, and CHOCO-SGD, need a diminishing or constant but small stepsize depending on the total number of iterations. However, these choices unavoidably cause slower convergence and bring in the difficulty of parameter tuning. In contrast, LEAD takes a different way to solve these issues, as explained below.

**Data heterogeneity**. It is common in distributed settings that there exists data heterogeneity among agents, especially in real-world applications where different agents collect data from different scenarios. In other words, we generally have $f_i(\boldsymbol{x}) \neq f_j(\boldsymbol{x})$ for $i \neq j$. The optimality condition of problem (1) gives $\mathbf{1}_{n\times 1}^\top \nabla\mathbf{F}(\mathbf{X}^*) = \mathbf{0}$, where $\mathbf{X}^* = [\boldsymbol{x}^*, \cdots, \boldsymbol{x}^*]$ is a consensual and optimal solution. The data heterogeneity and optimality condition imply that there exist at least two agents $i$ and $j$ such that $\nabla f_i(\boldsymbol{x}^*) \neq \mathbf{0}$ and $\nabla f_j(\boldsymbol{x}^*) \neq \mathbf{0}$. As a result, a simple D-PSGD algorithm cannot converge to the consensual and optimal solution as $\mathbf{X}^* \neq \mathbf{W}\mathbf{X}^* - \eta\mathbb{E}_\xi\nabla\mathbf{F}(\mathbf{X}^*; \xi)$ even when the stochastic gradient variance is zero.

*Gradient correction.* Primal-dual algorithms or gradient tracking algorithms are able to convergence much faster than DGD-type algorithms by handling the data heterogeneity issue, as introduced in Section 2. Specifically, LEAD is motivated by the design of primal-dual algorithm NIDS (Li et al., 2019) and the relation becomes clear if we consider the two-step reformulation of NIDS adopted in (Li & Yan, 2019):

$$\mathbf{D}^{k+1} = \mathbf{D}^k + \frac{\mathbf{I} - \mathbf{W}}{2\eta}(\mathbf{X}^k - \eta\nabla\mathbf{F}(\mathbf{X}^k) - \eta\mathbf{D}^k), \tag{4}$$

$$\mathbf{X}^{k+1} = \mathbf{X}^k - \eta\nabla\mathbf{F}(\mathbf{X}^k) - \eta\mathbf{D}^{k+1}, \tag{5}$$

where $\mathbf{X}^k$ and $\mathbf{D}^k$ represent the primal and dual variables respectively. The dual variable $\mathbf{D}^k$ plays the role of gradient correction. As $k \to \infty$, we expect $\mathbf{D}^k \to -\nabla\mathbf{F}(\mathbf{X}^*)$ and $\mathbf{X}^k$ will converge to $\mathbf{X}^*$ via the update in (5) since $\mathbf{D}^{k+1}$ corrects the nonzero gradient $\nabla\mathbf{F}(\mathbf{X}^k)$ asymptotically. The key design of Alg. 1 is to provide compression for the auxiliary variable defined as $\mathbf{Y}^k = \mathbf{X}^k - \eta\nabla\mathbf{F}(\mathbf{X}^k) - \eta\mathbf{D}^k$. Such design ensures that the dual variable $\mathbf{D}^k$ lies in **Range**$(\mathbf{I} - \mathbf{W})$, which is essential for convergence. Moreover, it achieves the implicit error compression as we will explain later. To stabilize the algorithm with inexact dual update, we introduce a parameter $\gamma$ to control the stepsize in the dual update. Therefore, if we ignore the details of the compression, Alg. 1 can be concisely written as

$$\mathbf{Y}^k = \mathbf{X}^k - \eta\nabla\mathbf{F}(\mathbf{X}^k; \xi^k) - \eta\mathbf{D}^k \tag{6}$$

$$\mathbf{D}^{k+1} = \mathbf{D}^k + \frac{\gamma}{2\eta}(\mathbf{I} - \mathbf{W})\hat{\mathbf{Y}}^k \tag{7}$$

$$\mathbf{X}^{k+1} = \mathbf{X}^k - \eta\nabla\mathbf{F}(\mathbf{X}^k; \xi^k) - \eta\mathbf{D}^{k+1} \tag{8}$$

where $\hat{\mathbf{Y}}^k$ represents the compression of $\mathbf{Y}^k$ and $\mathbf{F}(\mathbf{X}^k; \xi^k)$ denote the stochastic gradients.

Nevertheless, how to compress the communication and how fast the convergence we can attain with compression error are unknown. In the following, we propose to carefully control the compression error by difference compression and error compensation such that the inexact dual update (Line 6) and primal update (Line 7) can still guarantee the convergence as proved in Section 4.

**Compression error**. Different from existing works, which typically compress the primal variable $\mathbf{X}^k$ or its difference, LEAD first construct an intermediate variable $\mathbf{Y}^k$ and apply compression to obtain its coarse representation $\hat{\mathbf{Y}}^k$ as shown in the procedure COMM($\mathbf{Y}, \mathbf{H}, \mathbf{H}_w$):

- Compress the difference between $\mathbf{Y}$ and the state variable $\mathbf{H}$ as $\mathbf{Q}$;
- $\mathbf{Q}$ is encoded into the low-bit representation, which enables the efficient local communication step $\hat{\mathbf{Y}}_w = \mathbf{H}_w + \mathbf{W}\mathbf{Q}$. It is *the only communication step* in each iteration.
- Each agent recovers its estimate $\hat{\mathbf{Y}}$ by $\hat{\mathbf{Y}} = \mathbf{H} + \mathbf{Q}$ and we have $\hat{\mathbf{Y}}_w = \mathbf{W}\hat{\mathbf{Y}}$.
- States $\mathbf{H}$ and $\mathbf{H}_w$ are updated based on $\hat{\mathbf{Y}}$ and $\hat{\mathbf{Y}}_w$, respectively. We have $\mathbf{H}_w = \mathbf{W}\mathbf{H}$.

By this procedure, we expect when both $\mathbf{Y}^k$ and $\mathbf{H}^k$ converge to $\mathbf{X}^*$, the compression error vanishes asymptotically due to the assumption we make for the compression operator in Assumption 2.

**Remark 1.** *Note that difference compression is also applied in DCD-PSGD (Tang et al., 2018a) and CHOCO-SGD (Koloskova et al., 2019), but their state update is the simple integration of the compressed difference. We find this update is usually too aggressive and cause instability as showed in our experiments. Therefore, we adopt a momentum update $\mathbf{H} = (1 - \alpha)\mathbf{H} + \alpha\hat{\mathbf{Y}}$ motivated from DIANA (Mishchenko et al., 2019), which reduces the compression error for gradient compression in centralized optimization.*

*Implicit error compensation.* On the other hand, even if the compression error exists, LEAD essentially compensates for the error in the inexact dual update (Line 6), making the algorithm more stable and robust. To illustrate how it works, let $\mathbf{E}^k = \hat{\mathbf{Y}}^k - \mathbf{Y}^k$ denote the compression error and $e_i^k$ be its $i$-th row. The update of $\mathbf{D}^k$ gives

$$\mathbf{D}^{k+1} = \mathbf{D}^k + \frac{\gamma}{2\eta}(\hat{\mathbf{Y}}^k - \hat{\mathbf{Y}}_w^k) = \mathbf{D}^k + \frac{\gamma}{2\eta}(\mathbf{I} - \mathbf{W})\mathbf{Y}^k + \frac{\gamma}{2\eta}(\mathbf{E}^k - \mathbf{W}\mathbf{E}^k)$$

where $-\mathbf{W}\mathbf{E}^k$ indicates that agent $i$ spreads total compression error $-\sum_{j \in \mathcal{N}_i \cup \{i\}} w_{ji} e_i^k = -e_i^k$ to all agents and $\mathbf{E}^k$ indicates that each agent compensates this error locally by adding $e_i^k$ back. This error compensation also explains why the global view in (3) doesn't involve compression error.

**Remark 2.** *Note that in LEAD, the compression error is compensated into the model $\mathbf{X}^{k+1}$ through Line 6 and Line 7 such that the gradient computation in the next iteration is aware of the compression error. This has some subtle but important difference from the error compensation or error feedback in (Seide et al., 2014; Wu et al., 2018; Stich et al., 2018; Karimireddy et al., 2019; Tang et al., 2019b; Liu et al., 2020; Tang et al., 2019a), where the error is stored in the memory and only compensated after gradient computation and before the compression.*

**Remark 3.** *The proposed algorithm, LEAD in Alg. 1, recovers NIDS (Li et al., 2019), $D^2$ (Tang et al., 2018b), Exact Diffusion (Yuan et al., 2018). These connections are established in Appendix B.*

## 4 THEORETICAL ANALYSIS

In this section, we show the convergence rate for the proposed algorithm LEAD. Before showing the main theorem, we make some assumptions, which are commonly used for the analysis of decentralized optimization algorithms. All proofs are provided in Appendix E.

**Assumption 2** (Unbiased and $C$-contracted operator). *The compression operator $Q : \mathbb{R}^d \to \mathbb{R}^d$ is unbiased, i.e., $\mathbb{E}Q(\boldsymbol{x}) = \boldsymbol{x}$, and there exists $C \geq 0$ such that $\mathbb{E}\|\boldsymbol{x} - Q(\boldsymbol{x})\|_2^2 \leq C\|\boldsymbol{x}\|_2^2$ for all $\boldsymbol{x} \in \mathbb{R}^d$.*

**Assumption 3** (Stochastic gradient). *The stochastic gradient $\nabla f_i(\boldsymbol{x}; \xi)$ is unbiased, i.e., $\mathbb{E}_\xi \nabla f_i(\boldsymbol{x}; \xi) = \nabla f_i(\boldsymbol{x})$, and the stochastic gradient variance is bounded: $\mathbb{E}_\xi \|\nabla f_i(\boldsymbol{x}; \xi) - \nabla f_i(\boldsymbol{x})\|_2^2 \leq \sigma_i^2$ for all $i \in [n]$. Denote $\sigma^2 = \frac{1}{n}\sum_{i=1}^n \sigma_i^2$.*

**Assumption 4.** *Each $f_i$ is $L$-smooth and $\mu$-strongly convex with $L \geq \mu > 0$, i.e., for $i = 1, 2, \ldots, n$ and $\forall \boldsymbol{x}, \boldsymbol{y} \in \mathbb{R}^d$, we have*

$$f_i(\boldsymbol{y}) + \langle \nabla f_i(\boldsymbol{y}), \boldsymbol{x} - \boldsymbol{y} \rangle + \frac{\mu}{2}\|\boldsymbol{x} - \boldsymbol{y}\|^2 \leq f_i(\boldsymbol{x}) \leq f_i(\boldsymbol{y}) + \langle \nabla f_i(\boldsymbol{y}), \boldsymbol{x} - \boldsymbol{y} \rangle + \frac{L}{2}\|\boldsymbol{x} - \boldsymbol{y}\|^2.$$

**Theorem 1** (Constant stepsize). *Let $\{\mathbf{X}^k, \mathbf{H}^k, \mathbf{D}^k\}$ be the sequence generated from Alg. 1 and $\mathbf{X}^*$ is the optimal solution with $\mathbf{D}^* = -\nabla \mathbf{F}(\mathbf{X}^*)$. Under Assumptions 1-4, for any constant stepsize $\eta \in (0, 2/(\mu + L)]$, if the compression parameters $\alpha$ and $\gamma$ satisfy*

$$\gamma \in \left(0, \min\left\{\frac{2}{(3C+1)\beta}, \frac{2\mu\eta(2-\mu\eta)}{[2-\mu\eta(2-\mu\eta)]C\beta}\right\}\right), \tag{9}$$

$$\alpha \in \left[\frac{C\beta\gamma}{2(1+C)}, \frac{1}{a_1}\min\left\{\frac{2-\beta\gamma}{4-\beta\gamma}, \mu\eta(2-\mu\eta)\right\}\right], \tag{10}$$

*with $\beta := \lambda_{\max}(\mathbf{I} - \mathbf{W})$. Then, in total expectation we have*

$$\frac{1}{n}\mathbb{E}\mathcal{L}^{k+1} \leq \rho\frac{1}{n}\mathbb{E}\mathcal{L}^k + \eta^2\sigma^2, \tag{11}$$

*where*

$$\mathcal{L}^k := (1 - a_1\alpha)\|\mathbf{X}^k - \mathbf{X}^*\|^2 + (2\eta^2/\gamma)\mathbb{E}\|\mathbf{D}^k - \mathbf{D}^*\|^2_{(\mathbf{I}-\mathbf{W})^\dagger} + a_1\|\mathbf{H}^k - \mathbf{X}^*\|^2,$$

$$\rho := \max\left\{\frac{1 - \mu\eta(2-\mu\eta)}{1 - a_1\alpha}, 1 - \frac{\gamma}{2\lambda_{\max}((\mathbf{I}-\mathbf{W})^\dagger)}, \; 1 - \alpha\right\} < 1, a_1 := \frac{4(1+C)}{C\beta\gamma + 2}$$

*The result holds for $C \to 0$.*

**Corollary 1** (Complexity bounds). *Define the condition numbers of the objective function and communication graph as $\kappa_f = \frac{L}{\mu}$ and $\kappa_g = \frac{\lambda_{\max}(\mathbf{I}-\mathbf{W})}{\lambda^+_{\min}(\mathbf{I}-\mathbf{W})}$, respectively. Under the same setting in Theorem 1, we can choose $\eta = \frac{1}{L}, \gamma = \min\{\frac{1}{C\beta\kappa_f}, \frac{1}{(1+3C)\beta}\}$, and $\alpha = \mathcal{O}(\frac{1}{(1+C)\kappa_f})$ such that*

$$\rho = \max\left\{1 - \mathcal{O}\left(\frac{1}{(1+C)\kappa_f}\right), 1 - \mathcal{O}\left(\frac{1}{(1+C)\kappa_g}\right), 1 - \mathcal{O}\left(\frac{1}{C\kappa_f\kappa_g}\right)\right\}.$$

*With full-gradient (i.e., $\sigma = 0$), we obtain the following complexity bounds:*

- *LEAD converges to the $\epsilon$-accurate solution with the iteration complexity*

$$\mathcal{O}\left(\left((1+C)(\kappa_f + \kappa_g) + C\kappa_f\kappa_g\right)\log\frac{1}{\epsilon}\right).$$

- *When $C = 0$ (i.e., there is no compression), we obtain $\rho = \max\{1 - \mathcal{O}(\frac{1}{\kappa_f}), 1 - \mathcal{O}(\frac{1}{\kappa_g})\}$, and the iteration complexity $\mathcal{O}\left((\kappa_f + \kappa_g)\log\frac{1}{\epsilon}\right)$. This exactly recovers the convergence rate of NIDS (Li et al., 2019).*

- *When $C \leq \frac{\kappa_f + \kappa_g}{\kappa_f\kappa_g + \kappa_f + \kappa_g}$, the asymptotical complexity is $\mathcal{O}\left((\kappa_f + \kappa_g)\log\frac{1}{\epsilon}\right)$, which also recovers that of NIDS (Li et al., 2019) and indicates that the compression doesn't harm the convergence in this case.*

- *With $C = 0$ (or $C \leq \frac{\kappa_f + \kappa_g}{\kappa_f\kappa_g + \kappa_f + \kappa_g}$) and fully connected communication graph (i.e., $\mathbf{W} = \frac{\mathbf{1}\mathbf{1}^\top}{n}$), we have $\beta = 1$ and $\kappa_g = 1$. Therefore, we obtain $\rho = 1 - \mathcal{O}(\frac{1}{\kappa_f})$ and the complexity bound $\mathcal{O}(\kappa_f log\frac{1}{\epsilon})$. This recovers the convergence rate of gradient descent (Nesterov, 2013).*

**Remark 4.** *Under the setting in Theorem 1, LEAD converges linearly to the $\mathcal{O}(\sigma^2)$ neighborhood of the optimum and converges linearly exactly to the optimum if full gradient is used, e.g., $\sigma = 0$. The linear convergence of LEAD holds when $\eta < 2/L$, but we omit the proof.*

**Remark 5** (Arbitrary compression precision). *Pick any $\eta \in (0, 2/(\mu + L)]$, based on the compression-related constant $C$ and the network-related constant $\beta$, we can select $\gamma$ and $\alpha$ in certain ranges to achieve the convergence. It suggests that LEAD supports unbiased compression with arbitrary precision, i.e., any $C > 0$.*

**Corollary 2** (Consensus error). *Under the same setting in Theorem 1 , let $\bar{\boldsymbol{x}}^k = \frac{1}{n}\sum_{i=1}^n \boldsymbol{x}_i^k$ be the averaged model and $\mathbf{H}^0 = \mathbf{H}^1$, then all agents achieve consensus at the rate*

$$\frac{1}{n}\sum_{i=1}^n \mathbb{E}\left\|\boldsymbol{x}_i^k - \bar{\boldsymbol{x}}^k\right\|^2 \leq \frac{2\mathcal{L}^0}{n}\rho^k + \frac{2\sigma^2}{1-\rho}\eta^2. \tag{12}$$

*where $\rho$ is defined as in Corollary 1 with appropriate parameter settings.*

**Theorem 2** (Diminishing stepsize). *Let $\{\mathbf{X}^k, \mathbf{H}^k, \mathbf{D}^k\}$ be the sequence generated from Alg. 1 and $\mathbf{X}^*$ is the optimal solution with $\mathbf{D}^* = -\nabla\mathbf{F}(\mathbf{X}^*)$. Under Assumptions 1-4, if $\eta_k = \frac{2\theta_5}{\theta_3\theta_4\theta_5 k+2}$ and $\gamma_k = \theta_4\eta_k$, by taking $\alpha_k = \frac{C\beta\gamma_k}{2(1+C)}$, in total expectation we have*

$$\frac{1}{n}\sum_{i=1}^n \mathbb{E}\left\|\boldsymbol{x}_i^k - \boldsymbol{x}^*\right\|^2 \lesssim \mathcal{O}\left(\frac{1}{k}\right) \tag{13}$$

*where $\theta_1, \theta_2, \theta_3, \theta_4$ and $\theta_5$ are constants defined in the proof. The complexity bound for arriving at the $\epsilon$-accurate solution is $\mathcal{O}(\frac{1}{\epsilon})$.*

**Remark 6.** *Compared with CHOCO-SGD, LEAD requires unbiased compression and the convergence under biased compression is not investigated yet. The analysis of CHOCO-SGD relies on the bounded gradient assumptions, i.e., $\|\nabla f_i(\boldsymbol{x})\|^2 \leq G$, which is restrictive because it conflicts with the strong convexity while LEAD doesn't need this assumption. Moreover, in the theorem of CHOCO-SGD, it requires a specific point set of $\gamma$ while LEAD only requires $\gamma$ to be within a rather large range. This may explain the advantages of LEAD over CHOCO-SGD in terms of robustness to parameter setting.*

## 5 NUMERICAL EXPERIMENT

We consider three machine learning problems – $\ell_2$-regularized linear regression, logistic regression, and deep neural network. The proposed LEAD is compared with QDGD (Reisizadeh et al., 2019a), DeepSqueeze (Tang et al., 2019a), CHOCO-SGD (Koloskova et al., 2019), and two non-compressed algorithms DGD (Yuan et al., 2016) and NIDS (Li et al., 2019).

**Setup.** We consider eight machines connected in a ring topology network. Each agent can only exchange information with its two 1-hop neighbors. The mixing weight is simply set as $1/3$. For compression, we use the unbiased $b$-bits quantization method with $\infty$-norm

$$Q_\infty(\boldsymbol{x}) := \left(\|\boldsymbol{x}\|_\infty 2^{-(b-1)}\operatorname{sign}(\boldsymbol{x})\right) \cdot \left\lfloor \frac{2^{(b-1)}|\boldsymbol{x}|}{\|\boldsymbol{x}\|_\infty} + \boldsymbol{u} \right\rfloor, \tag{14}$$

where $\cdot$ is the Hadamard product, $|\boldsymbol{x}|$ is the elementwise absolute value of $\boldsymbol{x}$, and $\boldsymbol{u}$ is a random vector uniformly distributed in $[0, 1]^d$. Only $\operatorname{sign}(\boldsymbol{x})$, norm $\|\boldsymbol{x}\|_\infty$, and integers in the bracket need to be transmitted. Note that this quantization method is similar to the quantization used in QSGD (Alistarh et al., 2017) and CHOCO-SGD (Koloskova et al., 2019), but we use the $\infty$-norm scaling instead of the 2-norm. This small change brings significant improvement on compression precision as justified both theoretically and empirically in Appendix C. In this section, we choose 2-bit quantization and quantize the data blockwise (block size = 512).

For all experiments, we tune the stepsize $\eta$ from $\{0.01, 0.05, 0.1, 0.5\}$. For QDGD, CHOCO-SGD and Deepsqueeze, $\gamma$ is tuned from $\{0.01, 0.1, 0.2, 0.4, 0.6, 0.8, 1.0\}$. Note that different notations are used in their original papers. Here we uniformly denote the stepsize as $\eta$ and the additional parameter in these algorithms as $\gamma$ for simplicity. For LEAD, we simply fix $\alpha = 0.5$ and $\gamma = 1.0$ for all experiments since we find LEAD is robust to parameter settings as we validate in the parameter sensitivity analysis in Appendix D.1. This indicates the minor effort needed for tuning LEAD. Detailed parameter settings for all experiments are summarized in Appendix D.3.

**Linear regression.** We consider the problem: $f(\boldsymbol{x}) = \sum_{i=1}^n (\|\mathbf{A}_i\boldsymbol{x} - \boldsymbol{b}_i\|^2 + \lambda\|\boldsymbol{x}\|^2)$. Data matrices $\mathbf{A}_i \in \mathbb{R}^{200\times200}$ and the true solution $\boldsymbol{x}'$ is randomly synthesized. The values $\boldsymbol{b}_i$ are generated by adding Gaussian noise to $\mathbf{A}_i\boldsymbol{x}'$. We let $\lambda = 0.1$ and the optimal solution of the linear regression problem be $\boldsymbol{x}^*$. We use full-batch gradient to exclude the impact of gradient variance. The performance is showed in Fig. 1. The distance to $\boldsymbol{x}^*$ in Fig. 1a and the consensus error in Fig. 1c verify

that LEAD converges exponentially to the optimal consensual solution. It significantly outperforms most baselines and matches NIDS well under the same number of iterations. Fig. 1b demonstrates the benefit of compression when considering the communication bits. Fig. 1d shows that the compression error vanishes for both LEAD and CHOCO-SGD while the compression error is pretty large for QDGD and DeepSqueeze because they directly compress the local models.

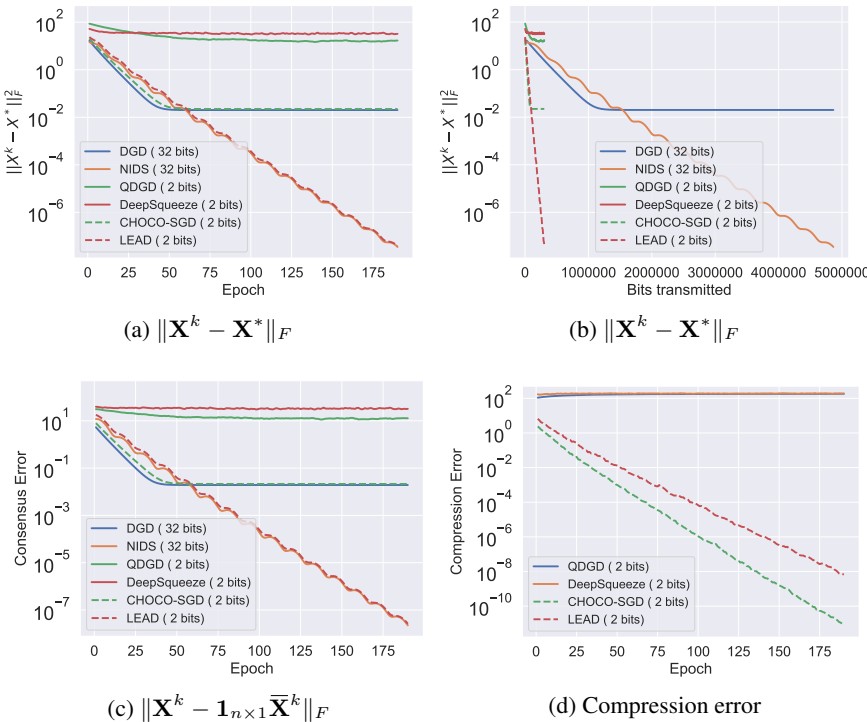

(a) $\|\mathbf{X}^k - \mathbf{X}^*\|_F$

(b) $\|\mathbf{X}^k - \mathbf{X}^*\|_F$

(c) $\|\mathbf{X}^k - \mathbf{1}_{n \times 1}\overline{\mathbf{X}}^k\|_F$

(d) Compression error

Figure 1: Linear regression problem.

**Logistic regression.** We further consider a logistic regression problem on the MNIST dataset. The regularization parameter is $10^{-4}$. We consider both *homogeneous* and *heterogeneous* data settings. In the homogeneous setting, the data samples are randomly shuffled before being uniformly partitioned among all agents such that the data distribution from each agent is very similar. In the heterogeneous setting, the samples are first sorted by their labels and then partitioned among agents. Due to the space limit, we mainly present the results in heterogeneous setting here and defer the homogeneous setting to Appendix D.2. The results using full-batch gradient and mini-batch gradient (the mini-batch size is $512$ for each agent) are showed in Fig. 2 and Fig. 3 respectively and both settings shows the faster convergence and higher precision of LEAD.

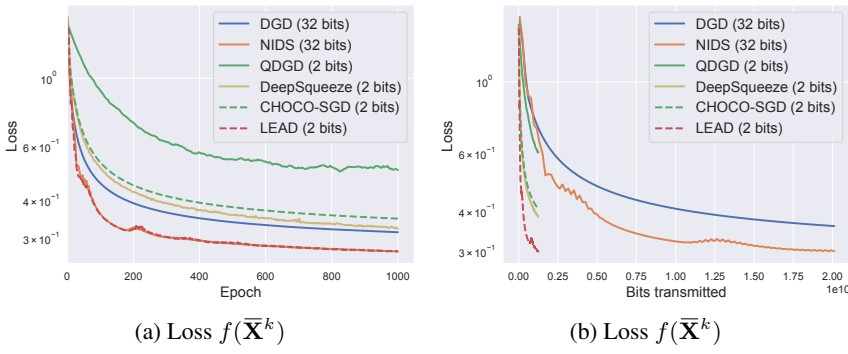

(a) Loss $f(\overline{\mathbf{X}}^k)$

(b) Loss $f(\overline{\mathbf{X}}^k)$

Figure 2: Logistic regression problem in the heterogeneous case (full-batch gradient).

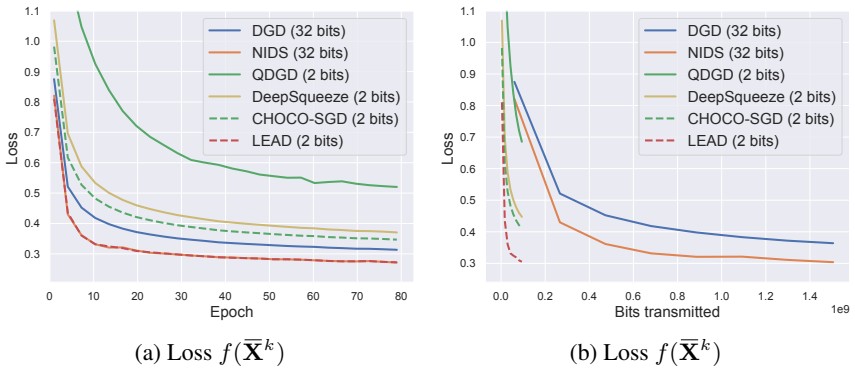

(a) Loss $f(\overline{\mathbf{X}}^k)$        (b) Loss $f(\overline{\mathbf{X}}^k)$

Figure 3: Logistic regression in the heterogeneous case (mini-batch gradient).

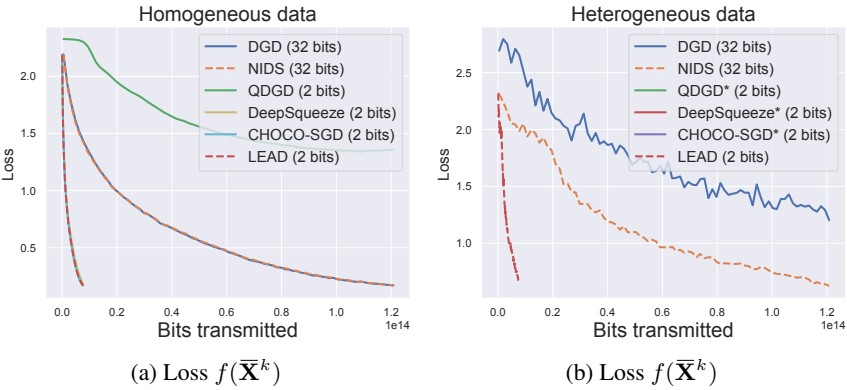

(a) Loss $f(\overline{\mathbf{X}}^k)$        (b) Loss $f(\overline{\mathbf{X}}^k)$

Figure 4: Stochastic optimization on deep neural network ($*$ means divergence).

**Neural network.** We empirically study the performance of LEAD in optimizing deep neural network by training AlexNet ($240$ MB) on CIFAR10 dataset. The mini-batch size is $64$ for each agents. Both the homogeneous and heterogeneous case are showed in Fig. 4. In the homogeneous case, CHOCO-SGD, DeepSqueeze and LEAD perform similarly and outperform the non-compressed variants in terms of communication efficiency, but CHOCO-SGD and DeepSqueeze need more efforts for parameter tuning because their convergence is sensitive to the setting of $\gamma$. In the heterogeneous cases, LEAD achieves the fastest and most stable convergence. Note that in this setting, sufficient information exchange is more important for convergence because models from different agents are moving to significantly diverse directions. In such case, DGD only converges with smaller stepsize and its communication compressed variants, including QDGD, DeepSqueeze and CHOCO-SGD, diverge in all parameter settings we try.

In summary, our experiments verify our theoretical analysis and show that LEAD is able to handle data heterogeneity very well. Furthermore, the performance of LEAD is robust to parameter settings and needs less effort for parameter tuning, which is critical in real-world applications.

## 6 CONCLUSION

In this paper, we investigate the communication compression in decentralized optimization. Motivated by primal-dual algorithms, a novel decentralized algorithm with compression, LEAD, is proposed to achieve faster convergence rate and to better handle heterogeneous data while enjoying the benefit of efficient communication. The nontrivial analyses on the coupled dynamics of inexact primal and dual updates as well as compression error establish the linear convergence of LEAD when full gradient is used and the linear convergence to the $\mathcal{O}(\sigma^2)$ neighborhood of the optimum when stochastic gradient is used. Extensive experiments validate the theoretical analysis and demonstrate the state-of-the-art efficiency and robustness of LEAD. LEAD is also applicable to non-convex problems as empirically verified in the neural network experiments but we leave the non-convex analysis as the future work.

ACKNOWLEDGEMENTS

Xiaorui Liu and Dr. Jiliang Tang are supported by the National Science Foundation (NSF) under grant numbers CNS-1815636, IIS-1928278, IIS-1714741, IIS-1845081, IIS-1907704, and IIS-1955285. Yao Li and Dr. Ming Yan are supported by NSF grant DMS-2012439 and Facebook Faculty Research Award (Systems for ML). Dr. Rongrong Wang is supported by NSF grant CCF-1909523.

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

# Contents of Appendix

## A  LEAD IN AGENT'S PERSPECTIVE

In the main paper, we described the algorithm with matrix notations for concision. Here we further provide a complete algorithm description from the agents' perspective.

---

**Algorithm 2** LEAD in Agent's Perspective

---

**input:** stepsize $\eta$, compression parameters $(\alpha, \gamma)$, initial values $\boldsymbol{x}_i^0, \boldsymbol{h}_i^1, \boldsymbol{z}_i, \forall i \in \{1, 2, \ldots, n\}$

**output:** $\boldsymbol{x}_i^K, \forall i \in \{1, 2, \ldots, n\}$ or $\frac{\sum_{i=1}^n \boldsymbol{x}_i^K}{n}$

1: **for** each agent $i \in \{1, 2, \ldots, n\}$ **do**

2:      $\boldsymbol{d}_i^1 = \boldsymbol{z}_i - \sum_{j \in \mathcal{N}_i \cup \{i\}} w_{ij} \boldsymbol{z}_j$

3:      $(\boldsymbol{h}_w)_i^1 = \sum_{j \in \mathcal{N}_i \cup \{i\}} w_{ij} (\boldsymbol{h}_w)_j^1$

4:      $\boldsymbol{x}_i^1 = \boldsymbol{x}_i^0 - \eta \nabla f_i(\boldsymbol{x}_i^0; \xi_i^0)$

5: **end for**

6: **for** $k = 1, 2, \ldots, K - 1$ **do** in parallel for all agents $i \in \{1, 2, \ldots, n\}$

7:      compute $\nabla f_i(\boldsymbol{x}_i^k; \xi_i^k)$                                  ▷ Gradient computation

8:      $\boldsymbol{y}_i^k = \boldsymbol{x}_i^k - \eta \nabla f_i(\boldsymbol{x}_i^k; \xi_i^k) - \eta \boldsymbol{d}_i^k$

9:      $\boldsymbol{q}_i^k = \text{Compress}(\boldsymbol{y}_i^k - \boldsymbol{h}_i^k)$                         ▷ Compression

10:      $\hat{\boldsymbol{y}}_i^k = \boldsymbol{h}_i^k + \boldsymbol{q}_i^k$

11:      **for** neighbors $j \in \mathcal{N}_i$ **do**

12:          Send $\boldsymbol{q}_i^k$ and receive $\boldsymbol{q}_j^k$                      ▷ Communication

13:      **end for**

14:      $(\hat{\boldsymbol{y}}_w)_i^k = (\boldsymbol{h}_w)_i^k + \sum_{j \in \mathcal{N}_i \cup \{i\}} w_{ij} \boldsymbol{q}_j^k$

15:      $\boldsymbol{h}_i^{k+1} = (1 - \alpha) \boldsymbol{h}_i^k + \alpha \hat{\boldsymbol{y}}_i^k$

16:      $(\boldsymbol{h}_w)_i^{k+1} = (1 - \alpha)(\boldsymbol{h}_w)_i^k + \alpha (\hat{\boldsymbol{y}}_w)_i^k$

17:      $\boldsymbol{d}_i^{k+1} = \boldsymbol{d}_i^k + \frac{\gamma}{2\eta} (\hat{\boldsymbol{y}}_i^k - (\hat{\boldsymbol{y}}_w)_i^k)$

18:      $\boldsymbol{x}_i^{k+1} = \boldsymbol{x}_i^k - \eta \nabla f_i(\boldsymbol{x}_i^k; \xi_i^k) - \eta \boldsymbol{d}_i^{k+1}$           ▷ Model update

19: **end for**

---

## B  CONNECTIONS WITH EXITING WORKS

The non-compressed variant of LEAD in Alg. 1 recovers NIDS (Li et al., 2019), $D^2$ (Tang et al., 2018b) and Exact Diffusion (Yuan et al., 2018) as shown in Proposition 1. In Corollary 3, we show that the convergence rate of LEAD exactly recovers the rate of NIDS when $C = 0, \gamma = 1$ and $\sigma = 0$.

**Proposition 1** (Connection to NIDS, $D^2$ and Exact Diffusion). *When there is no communication compression (i.e., $\hat{\mathbf{Y}}^k = \mathbf{Y}^k$) and $\gamma = 1$, Alg. 1 recovers $D^2$:*

$$\mathbf{X}^{k+1} = \frac{\mathbf{I} + \mathbf{W}}{2} \Big( 2\mathbf{X}^k - \mathbf{X}^{k-1} - \eta \nabla \mathbf{F}(\mathbf{X}^k; \xi^k) + \eta \nabla \mathbf{F}(\mathbf{X}^{k-1}; \xi^{k-1}) \Big). \tag{15}$$

*Furthermore, if the stochastic estimator of the gradient $\nabla \mathbf{F}(\mathbf{X}^k; \xi^k)$ is replaced by the full gradient, it recovers NIDS and Exact Diffusion with specific settings.*

**Corollary 3** (Consistency with NIDS). *When $C = 0$ (no communication compression), $\gamma = 1$ and $\sigma = 0$ (full gradient), LEAD has the convergence consistent with NIDS with $\eta \in (0, 2/(\mu + L)]$:*

$$\mathcal{L}^{k+1} \leq \max \left\{ 1 - \mu(2\eta - \mu\eta^2), 1 - \frac{1}{2\lambda_{\max}((\mathbf{I} - \mathbf{W})^\dagger)} \right\} \mathcal{L}^k. \tag{16}$$

See the proof in E.5.

*Proof of Proposition 1.* Let $\gamma = 1$ and $\hat{\mathbf{Y}}^k = \mathbf{Y}^k$. Combing Lines 4 and 6 of Alg. 1 gives

$$\mathbf{D}^{k+1} = \mathbf{D}^k + \frac{\mathbf{I} - \mathbf{W}}{2\eta} (\mathbf{X}^k - \eta \nabla \mathbf{F}(\mathbf{X}^k; \xi^k) - \eta \mathbf{D}^k). \tag{17}$$

Based on Line 7, we can represent $\eta \mathbf{D}^k$ from the previous iteration as

$$\eta \mathbf{D}^k = \mathbf{X}^{k-1} - \mathbf{X}^k - \eta \nabla \mathbf{F}(\mathbf{X}^{k-1}; \xi^{k-1}). \tag{18}$$

Eliminating both $\mathbf{D}^k$ and $\mathbf{D}^{k+1}$ by substituting (17)-(18) into Line 7, we obtain

$$
\begin{aligned}
\mathbf{X}^{k+1} &= \mathbf{X}^k - \eta \nabla \mathbf{F}(\mathbf{X}^k; \xi^k) - \left( \eta \mathbf{D}^k + \frac{\mathbf{I} - \mathbf{W}}{2}(\mathbf{X}^k - \eta \nabla \mathbf{F}(\mathbf{X}^k; \xi^k) - \eta \mathbf{D}^k) \right) \quad \text{(from (17))} \\
&= \frac{\mathbf{I} + \mathbf{W}}{2}(\mathbf{X}^k - \eta \nabla \mathbf{F}(\mathbf{X}^k; \xi^k)) - \frac{\mathbf{I} + \mathbf{W}}{2} \eta \mathbf{D}^k \\
&= \frac{\mathbf{I} + \mathbf{W}}{2}(\mathbf{X}^k - \eta \nabla \mathbf{F}(\mathbf{X}^k; \xi^k)) - \frac{\mathbf{I} + \mathbf{W}}{2}(\mathbf{X}^{k-1} - \mathbf{X}^k - \eta \nabla \mathbf{F}(\mathbf{X}^{k-1}; \xi^{k-1})) \quad \text{(from (18))} \\
&= \frac{\mathbf{I} + \mathbf{W}}{2}(2\mathbf{X}^k - \mathbf{X}^{k-1} - \eta \nabla \mathbf{F}(\mathbf{X}^k; \xi^k) + \eta \nabla \mathbf{F}(\mathbf{X}^{k-1}; \xi^{k-1})), \tag{19}
\end{aligned}
$$

which is exactly $D^2$. It also recovers Exact Diffusion with $\mathbf{A} = \frac{\mathbf{I}+\mathbf{W}}{2}$ and $\mathbf{M} = \eta \mathbf{I}$ in Eq. (97) of (Yuan et al., 2018). $\qquad \square$

## C  COMPRESSION METHOD

### C.1  P-NORM B-BITS QUANTIZATION

**Theorem 3** (p-norm b-bit quantization). *Let us define the quantization operator as*

$$Q_p(\boldsymbol{x}) := \left( \|\boldsymbol{x}\|_p \operatorname{sign}(\boldsymbol{x}) 2^{-(b-1)} \right) \cdot \left\lfloor \frac{2^{b-1}|\boldsymbol{x}|}{\|\boldsymbol{x}\|_p} + \boldsymbol{u} \right\rfloor \tag{20}$$

*where $\cdot$ is the Hadamard product, $|\boldsymbol{x}|$ is the elementwise absolute value and $\boldsymbol{u}$ is a random dither vector uniformly distributed in $[0,1]^d$. $Q_p(\boldsymbol{x})$ is unbiased, i.e., $\mathbb{E}Q_p(\boldsymbol{x}) = \boldsymbol{x}$, and the compression variance is upper bounded by*

$$\mathbb{E}\|\boldsymbol{x} - Q_p(\boldsymbol{x})\|^2 \leq \frac{1}{4}\|\operatorname{sign}(\boldsymbol{x}) 2^{-(b-1)}\|^2 \|\boldsymbol{x}\|_p^2, \tag{21}$$

*which suggests that $\infty$-norm provides the smallest upper bound for the compression variance due to $\|\boldsymbol{x}\|_p \leq \|\boldsymbol{x}\|_q, \forall \boldsymbol{x}$ if $1 \leq q \leq p \leq \infty$.*

**Remark 7.** *For the compressor defined in (20), we have the following the compression constant*

$$C = \sup_{\boldsymbol{x}} \frac{\|\operatorname{sign}(\boldsymbol{x}) 2^{-(b-1)}\|^2 \|\boldsymbol{x}\|_p^2}{4\|\boldsymbol{x}\|^2}.$$

*Proof.* Let denote $\boldsymbol{v} = \|\boldsymbol{x}\|_p \operatorname{sign}(\boldsymbol{x}) 2^{-(b-1)}$, $s = \frac{2^{b-1}|\boldsymbol{x}|}{\|\boldsymbol{x}\|_p}$, $s_1 = \left\lfloor \frac{2^{b-1}|\boldsymbol{x}|}{\|\boldsymbol{x}\|_p} \right\rfloor$ and $s_2 = \left\lceil \frac{2^{b-1}|\boldsymbol{x}|}{\|\boldsymbol{x}\|_p} \right\rceil$. We can rewrite $\boldsymbol{x}$ as $\boldsymbol{x} = s \cdot \boldsymbol{v}$.

For any coordinate $i$ such that $s_i = (s_1)_i$, we have $Q_p(\boldsymbol{x}_i) = (s_1)_i \boldsymbol{v}_i$ with probability 1. Hence $\mathbb{E}Q_p(\boldsymbol{x})_i = s_i \boldsymbol{v}_i = \boldsymbol{x}_i$ and

$$\mathbb{E}(\boldsymbol{x}_i - Q_p(\boldsymbol{x})_i)^2 = (\boldsymbol{x}_i - s_i \boldsymbol{v}_i)^2 = 0.$$

For any coordinate $i$ such that $s_i \neq (s_1)_i$, we have $(s_2)_i - (s_1)_i = 1$ and $Q_p(\boldsymbol{x})_i$ satisfies

$$Q_p(\boldsymbol{x})_i = \begin{cases} (s_1)_i \boldsymbol{v}_i, & \text{w.p. } (s_2)_i - s_i, \\ (s_2)_i \boldsymbol{v}_i, & \text{w.p. } s_i - (s_1)_i. \end{cases}$$

Thus, we derive

$$\mathbb{E}Q_p(\boldsymbol{x})_i = \boldsymbol{v}_i (s_1)_i (s_2 - s)_i + \boldsymbol{v}_i (s_2)_i (s - s_1)_i = \boldsymbol{v}_i s_i (s_2 - s_1)_i = \boldsymbol{v}_i s_i = \boldsymbol{x}_i,$$

and

$$\mathbb{E}[\boldsymbol{x}_i - Q_p(\boldsymbol{x})_i]^2 = (\boldsymbol{x}_i - \boldsymbol{v}_i(s_1)_i)^2(s_2 - s)_i + (\boldsymbol{x}_i - \boldsymbol{v}_i(s_2)_i)^2(s - s_1)_i$$
$$= (s_2 - s_1)_i\boldsymbol{x}_i^2 + \big((s_1)_i(s_2)_i(s_1 - s_2)_i + s_i((s_2)_i^2 - (s_1)_i^2)\big)\boldsymbol{v}_i^2 - 2s_i(s_2 - s_1)_i\boldsymbol{x}_i\boldsymbol{v}_i$$
$$= \boldsymbol{x}_i^2 + \big(-(s_1)_i(s_2)_i + s_i(s_2 + s_1)_i\big)\boldsymbol{v}_i^2 - 2s_i\boldsymbol{x}_i\boldsymbol{v}_i$$
$$= (\boldsymbol{x}_i - s_i\boldsymbol{v}_i)^2 + \big(-(s_1)_i(s_2)_i + s_i(s_2 + s_1)_i - s_i^2\big)\boldsymbol{v}_i^2$$
$$= (\boldsymbol{x}_i - s_i\boldsymbol{v}_i)^2 + (s_2 - s)_i(s - s_1)_i\boldsymbol{v}_i^2$$
$$= (s_2 - s)_i(s - s_1)_i\boldsymbol{v}_i^2$$
$$\leq \frac{1}{4}\boldsymbol{v}_i^2.$$

Considering both cases, we have $\mathbb{E}Q(\boldsymbol{x}) = \boldsymbol{x}$ and

$$\mathbb{E}\|\boldsymbol{x} - Q_p(\boldsymbol{x})\|^2 = \sum_{\{s_i = (s_1)_i\}} \mathbb{E}[\boldsymbol{x}_i - Q_p(\boldsymbol{x})_i]^2 + \sum_{\{s_i \neq (s_1)_i\}} \mathbb{E}[\boldsymbol{x}_i - Q_p(\boldsymbol{x})_i]^2$$
$$\leq 0 + \frac{1}{4}\sum_{\{s_i \neq (s_1)_i\}} \boldsymbol{v}_i^2$$
$$\leq \frac{1}{4}\|\boldsymbol{v}\|^2$$
$$= \frac{1}{4}\|\operatorname{sign}(\boldsymbol{x})2^{-(b-1)}\|^2\|\boldsymbol{x}\|_p^2.$$

$\square$

## C.2 COMPRESSION ERROR

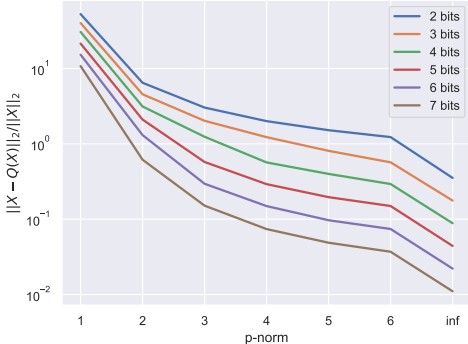

Figure 5: Relative compression error $\frac{\|\boldsymbol{x} - Q(\boldsymbol{x})\|_2}{\|\boldsymbol{x}\|_2}$ for p-norm b-bit quantization

To verify Theorem 3, we compare the compression error of the quantization method defined in (20) with different norms ($p = 1, 2, 3, \ldots, 6, \infty$). Specifically, we uniformly generate 100 random vectors in $\mathbb{R}^{10000}$ and compute the average compression error. The result shown in Figure 5 verifies our proof in Theorem 3 that the compression error decreases when $p$ increases. This suggests that $\infty$-norm provides the best compression precision under the same bit constraint.

Under similar setting, we also compare the compression error with other popular compression methods, such as top-k and random-k sparsification. The x-axes represents the average bits needed to represent each element of the vector. The result is showed in Fig. 6. Note that intuitively top-k methods should perform better than random-k method, but the top-k method needs extra bits to transmitted the index while random-k method can avoid this by using the same random seed. Therefore, top-k method doesn't outperform random-k too much under the same communication budget. The result in Fig. 6 suggests that $\infty$-norm b-bits quantization provides significantly better compression precision than others under the same bit constraint.

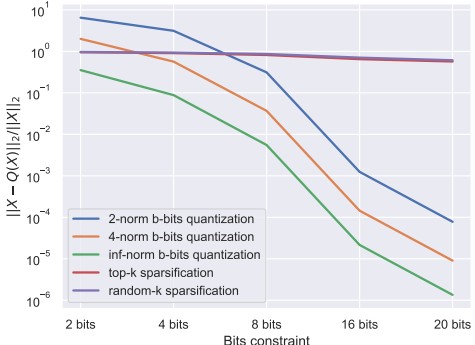

Figure 6: Comparison of compression error $\frac{\|\boldsymbol{x}-Q(\boldsymbol{x})\|_2}{\|\boldsymbol{x}\|_2}$ between different compression methods

# D EXPERIMENTS

## D.1 PARAMETER SENSITIVITY

In the linear regression problem, the convergence of LEAD under different parameter settings of $\alpha$ and $\gamma$ are tested. The result showed in Figure 7 indicates that LEAD performs well in most settings and is robust to the parameter setting. Therefore, in this paper, we simply set $\alpha = 0.5$ and $\gamma = 1.0$ for LEAD in all experiment, which indicates the minor effort needed for parameter tuning.

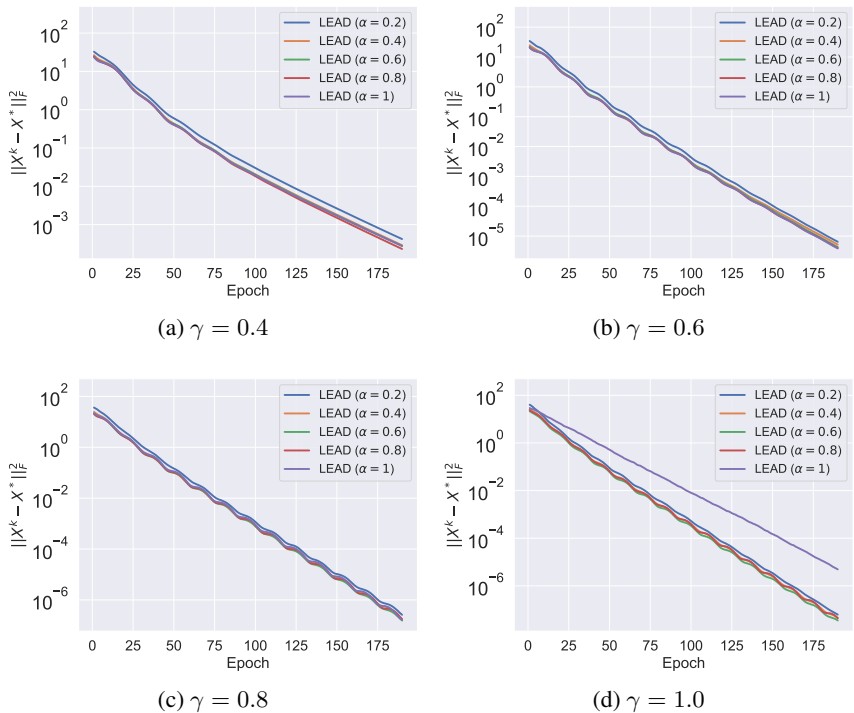

Figure 7: Parameter analysis on linear regression problem.

## D.2 EXPERIMENTS IN HOMOGENEOUS SETTING

The experiments on logistic regression problem in homogeneous case are showed in Fig. 8 and Fig. 9. It shows that DeepSqueeze, CHOCO-SGD and LEAD converges similarly while Deep-

Squeeze and CHOCO-SGD require to tune a smaller $\gamma$ for convergence as showed in the parameter setting in Section D.3. Generally, a smaller $\gamma$ decreases the model propagation between agents since $\gamma$ changes the effective mixing matrix and this may cause slower convergence. However, in the setting where data from different agents are very similar, the models move to close directions such that the convergence is not affected too much.

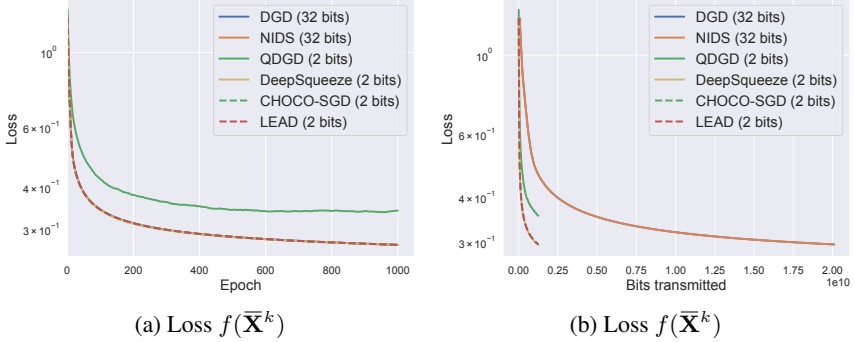

(a) Loss $f(\overline{\mathbf{X}}^k)$          (b) Loss $f(\overline{\mathbf{X}}^k)$

Figure 8: Logistic regression in the homogeneous case (full-batch gradient)

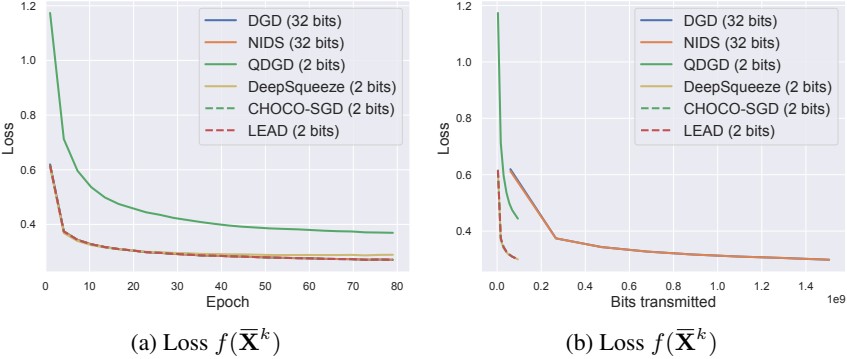

(a) Loss $f(\overline{\mathbf{X}}^k)$          (b) Loss $f(\overline{\mathbf{X}}^k)$

Figure 9: Logistic regression in the homogeneous case (mini-batch gradient)

## D.3 PARAMETER SETTINGS

The best parameter settings we search for all algorithms and experiments are summarized in Tables 1– 4. QDGD and DeepSqueeze are more sensitive to $\gamma$ and CHOCO-SGD is slight more robust. LEAD is most robust to parameter settings and it works well for the setting $\alpha = 0.5$ and $\gamma = 1.0$ in all experiments in this paper.

| Algorithm | $\eta$ | $\gamma$ | $\alpha$ |
|-----------|--------|----------|----------|
| DGD | 0.1 | - | - |
| NIDS | 0.1 | - | - |
| QDGD | 0.1 | 0.2 | - |
| DeepSqueeze | 0.1 | 0.2 | - |
| CHOCO-SGD | 0.1 | 0.8 | - |
| LEAD | 0.1 | 1.0 | 0.5 |

Table 1: Parameter settings for the linear regression problem.

| Algorithm | $\eta$ | $\gamma$ | $\alpha$ |
|-----------|--------|----------|----------|
| DGD | 0.1 | - | - |
| NIDS | 0.1 | - | - |
| QDGD | 0.1 | 0.4 | - |
| DeepSqueeze | 0.1 | 0.4 | - |
| CHOCO-SGD | 0.1 | 0.6 | - |
| LEAD | 0.1 | 1.0 | 0.5 |

Homogeneous case

| Algorithm | $\eta$ | $\gamma$ | $\alpha$ |
|-----------|--------|----------|----------|
| DGD | 0.1 | - | - |
| NIDS | 0.1 | - | - |
| QDGD | 0.1 | 0.2 | - |
| DeepSqueeze | 0.1 | 0.6 | - |
| CHOCO-SGD | 0.1 | 0.6 | - |
| LEAD | 0.1 | 1.0 | 0.5 |

Heterogeneous case

Table 2: Parameter settings for the logistic regression problem (full-batch gradient).

| Algorithm | $\eta$ | $\gamma$ | $\alpha$ |
|-----------|--------|----------|----------|
| DGD | 0.1 | - | - |
| NIDS | 0.1 | - | - |
| QDGD | 0.05 | 0.2 | - |
| DeepSqueeze | 0.1 | 0.6 | - |
| CHOCO-SGD | 0.1 | 0.6 | - |
| LEAD | 0.1 | 1.0 | 0.5 |

Homogeneous case

| Algorithm | $\eta$ | $\gamma$ | $\alpha$ |
|-----------|--------|----------|----------|
| DGD | 0.1 | - | - |
| NIDS | 0.1 | - | - |
| QDGD | 0.05 | 0.2 | - |
| DeepSqueeze | 0.1 | 0.6 | - |
| CHOCO-SGD | 0.1 | 0.6 | - |
| LEAD | 0.1 | 1.0 | 0.5 |

Heterogeneous case

Table 3: Parameter settings for the logistic regression problem (mini-batch gradient).

| Algorithm | $\eta$ | $\gamma$ | $\alpha$ |
|-----------|--------|----------|----------|
| DGD | 0.1 | - | - |
| NIDS | 0.1 | - | - |
| QDGD | 0.05 | 0.1 | - |
| DeepSqueeze | 0.1 | 0.2 | - |
| CHOCO-SGD | 0.1 | 0.6 | - |
| LEAD | 0.1 | 1.0 | 0.5 |

Homogeneous case

| Algorithm | $\eta$ | $\gamma$ | $\alpha$ |
|-----------|--------|----------|----------|
| DGD | 0.05 | - | - |
| NIDS | 0.1 | - | - |
| QDGD | * | * | - |
| DeepSqueeze | * | * | - |
| CHOCO-SGD | * | * | - |
| LEAD | 0.1 | 1.0 | 0.5 |

Heterogeneous case

Table 4: Parameter settings for the deep neural network. (* means divergence for all options we try)

# E  PROOFS OF THE THEOREMS

## E.1  ILLUSTRATIVE FLOW

The following flow graph depicts the relation between iterative variables and clarifies the range of conditional expectation. $\{\mathcal{G}_k\}_{k=0}^{\infty}$ and $\{\mathcal{F}_k\}_{k=0}^{\infty}$ are two $\sigma-$algebras generated by the gradient sampling and the stochastic compression respectively. They satisfy

$$\mathcal{G}_0 \subset \mathcal{F}_0 \subset \mathcal{G}_1 \subset \mathcal{F}_1 \subset \cdots \subset \mathcal{G}_k \subset \mathcal{F}_k \subset \cdots$$

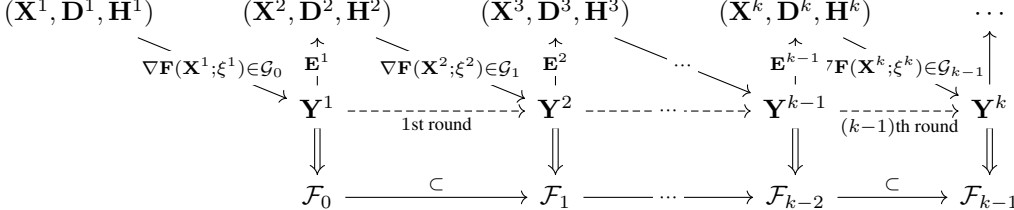

The solid and dashed arrows in the top flow illustrate the dynamics of the algorithm, while in the bottom, the arrows stand for the relation between successive $\mathcal{F}$-$\sigma$-algebras. The downward arrows

determine the range of $\mathcal{F}$-$\sigma$-algebras. E.g., up to $\mathbf{E}^k$, all random variables are in $\mathcal{F}_{k-1}$ and up to $\nabla\mathbf{F}(\mathbf{X}^k;\xi^k)$, all random variables are in $\mathcal{G}_{k-1}$ with $\mathcal{G}_{k-1} \subset \mathcal{F}_{k-1}$. Throughout the appendix, without specification, $\mathbb{E}$ is the expectation conditioned on the corresponding stochastic estimators given the context.

### E.2 Two central Lemmas

**Lemma 1** (Fundamental equality). *Let $\mathbf{X}^*$ be the optimal solution, $\mathbf{D}^* := -\nabla\mathbf{F}(\mathbf{X}^*)$ and $\mathbf{E}^k$ denote the compression error in the kth iteration, that is $\mathbf{E}^k = \mathbf{Q}^k - (\mathbf{Y}^k - \mathbf{H}^k) = \hat{\mathbf{Y}}^k - \mathbf{Y}^k$. From Alg. 1, we have*

$$
\begin{aligned}
&\|\mathbf{X}^{k+1} - \mathbf{X}^*\|^2 + (\eta^2/\gamma)\|\mathbf{D}^{k+1} - \mathbf{D}^*\|_{\mathbf{M}}^2 \\
=&\|\mathbf{X}^k - \mathbf{X}^*\|^2 + (\eta^2/\gamma)\|\mathbf{D}^k - \mathbf{D}^*\|_{\mathbf{M}}^2 - (\eta^2/\gamma)\|\mathbf{D}^{k+1} - \mathbf{D}^k\|_{\mathbf{M}}^2 - \eta^2\|\mathbf{D}^{k+1} - \mathbf{D}^*\|^2 \\
&- 2\eta\langle\mathbf{X}^k - \mathbf{X}^*, \nabla\mathbf{F}(\mathbf{X}^k;\xi^k) - \nabla\mathbf{F}(\mathbf{X}^*)\rangle + \eta^2\|\nabla\mathbf{F}(\mathbf{X}^k;\xi^k) - \nabla\mathbf{F}(\mathbf{X}^*)\|^2 + 2\eta\langle\mathbf{E}^k, \mathbf{D}^{k+1} - \mathbf{D}^*\rangle,
\end{aligned}
$$

*where $\mathbf{M} := 2(\mathbf{I} - \mathbf{W})^\dagger - \gamma\mathbf{I}$ and $\gamma < 2/\lambda_{\max}(\mathbf{I} - \mathbf{W})$ ensures the positive definiteness of $\mathbf{M}$ over $\mathrm{range}(\mathbf{I} - \mathbf{W})$.*

**Lemma 2** (State inequality). *Let the same assumptions in Lemma 1 hold. From Alg. 1, if we take the expectation over the compression operator conditioned on the $k$-th iteration, we have*

$$
\begin{aligned}
\mathbb{E}\|\mathbf{H}^{k+1} - \mathbf{X}^*\|^2 \le &(1-\alpha)\|\mathbf{H}^k - \mathbf{X}^*\|^2 + \alpha\mathbb{E}\|\mathbf{X}^{k+1} - \mathbf{X}^*\|^2 + \alpha\eta^2\mathbb{E}\|\mathbf{D}^{k+1} - \mathbf{D}^k\|^2 \\
&+ \frac{2\alpha\eta^2}{\gamma}\mathbb{E}\|\mathbf{D}^{k+1} - \mathbf{D}^k\|_{\mathbf{M}}^2 + \alpha^2\mathbb{E}\|\mathbf{E}^k\|^2 - \alpha\gamma\mathbb{E}\|\mathbf{E}^k\|_{\mathbf{I}-\mathbf{W}}^2 - \alpha(1-\alpha)\|\mathbf{Y}^k - \mathbf{H}^k\|^2.
\end{aligned}
$$

### E.3 Proof of Lemma 1

Before proving Lemma 1, we let $\mathbf{E}^k = \hat{\mathbf{Y}}^k - \mathbf{Y}^k$ and introduce the following three Lemmas.

**Lemma 3.** *Let $\mathbf{X}^*$ be the consensus solution. Then, from Line 4-7 of Alg. 1, we obtain*

$$
\frac{\mathbf{I} - \mathbf{W}}{2\eta}(\mathbf{X}^{k+1} - \mathbf{X}^*) = \left(\frac{I}{\gamma} - \frac{\mathbf{I} - \mathbf{W}}{2}\right)(\mathbf{D}^{k+1} - \mathbf{D}^k) - \frac{\mathbf{I} - \mathbf{W}}{2\eta}\mathbf{E}^k. \tag{22}
$$

*Proof.* From the iterations in Alg. 1, we have

$$
\begin{aligned}
\mathbf{D}^{k+1} &= \mathbf{D}^k + \frac{\gamma}{2\eta}(\mathbf{I} - \mathbf{W})\hat{\mathbf{Y}}^k \qquad \text{(from Line 6)} \\
&= \mathbf{D}^k + \frac{\gamma}{2\eta}(\mathbf{I} - \mathbf{W})(\mathbf{Y}^k + \mathbf{E}^k) \\
&= \mathbf{D}^k + \frac{\gamma}{2\eta}(\mathbf{I} - \mathbf{W})(\mathbf{X}^k - \eta\nabla\mathbf{F}(\mathbf{X}^k;\xi^k) - \eta\mathbf{D}^k + \mathbf{E}^k) \qquad \text{(from Line 4)} \\
&= \mathbf{D}^k + \frac{\gamma}{2\eta}(\mathbf{I} - \mathbf{W})(\mathbf{X}^k - \eta\nabla\mathbf{F}(\mathbf{X}^k;\xi^k) - \eta\mathbf{D}^{k+1} - \mathbf{X}^* + \eta(\mathbf{D}^{k+1} - \mathbf{D}^k) + \mathbf{E}^k) \\
&= \mathbf{D}^k + \frac{\gamma}{2\eta}(\mathbf{I} - \mathbf{W})(\mathbf{X}^{k+1} - \mathbf{X}^*) + \frac{\gamma}{2}(\mathbf{I} - \mathbf{W})(\mathbf{D}^{k+1} - \mathbf{D}^k) + \frac{\gamma}{2\eta}(\mathbf{I} - \mathbf{W})\mathbf{E}^k,
\end{aligned}
$$

where the fourth equality holds due to $(\mathbf{I} - \mathbf{W})\mathbf{X}^* = \mathbf{0}$ and the last equality comes from Line 7 of Alg. 1. Rewriting this equality, and we obtain (22). $\square$

**Lemma 4.** *Let $\mathbf{D}^* = -\nabla\mathbf{F}(\mathbf{X}^*) \in \mathrm{span}\{\mathbf{I} - \mathbf{W}\}$, we have*

$$
\langle\mathbf{X}^{k+1} - \mathbf{X}^*, \mathbf{D}^{k+1} - \mathbf{D}^k\rangle = \frac{\eta}{\gamma}\|\mathbf{D}^{k+1} - \mathbf{D}^k\|_{\mathbf{M}}^2 - \langle\mathbf{E}^k, \mathbf{D}^{k+1} - \mathbf{D}^k\rangle, \tag{23}
$$

$$
\langle\mathbf{X}^{k+1} - \mathbf{X}^*, \mathbf{D}^{k+1} - \mathbf{D}^*\rangle = \frac{\eta}{\gamma}\langle\mathbf{D}^{k+1} - \mathbf{D}^k, \mathbf{D}^{k+1} - \mathbf{D}^*\rangle_{\mathbf{M}} - \langle\mathbf{E}^k, \mathbf{D}^{k+1} - \mathbf{D}^*\rangle, \tag{24}
$$

*where $\mathbf{M} = 2(\mathbf{I} - \mathbf{W})^\dagger - \gamma\mathbf{I}$ and $\gamma < 2/\lambda_{\max}(\mathbf{I} - \mathbf{W})$ ensures the positive definiteness of $\mathbf{M}$ over $\mathrm{span}\{\mathbf{I} - \mathbf{W}\}$.*

*Proof.* Since $\mathbf{D}^{k+1} \in \mathbf{span}\{\mathbf{I} - \mathbf{W}\}$ for any $k$, we have

$$
\begin{aligned}
&\langle \mathbf{X}^{k+1} - \mathbf{X}^*, \mathbf{D}^{k+1} - \mathbf{D}^k \rangle \\
=&\langle (\mathbf{I} - \mathbf{W})(\mathbf{X}^{k+1} - \mathbf{X}^*), (\mathbf{I} - \mathbf{W})^\dagger (\mathbf{D}^{k+1} - \mathbf{D}^k) \rangle \\
=&\left\langle \frac{\eta}{\gamma}(2\mathbf{I} - \gamma(\mathbf{I} - \mathbf{W}))(\mathbf{D}^{k+1} - \mathbf{D}^k) - (\mathbf{I} - \mathbf{W})\mathbf{E}^k, (\mathbf{I} - \mathbf{W})^\dagger (\mathbf{D}^{k+1} - \mathbf{D}^k) \right\rangle \quad \text{(from (22))} \\
=&\left\langle \frac{\eta}{\gamma}(2(\mathbf{I} - \mathbf{W})^\dagger - \gamma\mathbf{I})(\mathbf{D}^{k+1} - \mathbf{D}^k) - \mathbf{E}^k, \mathbf{D}^{k+1} - \mathbf{D}^k \right\rangle \\
=&\frac{\eta}{\gamma}\|\mathbf{D}^{k+1} - \mathbf{D}^k\|_{\mathbf{M}}^2 - \langle \mathbf{E}^k, \mathbf{D}^{k+1} - \mathbf{D}^k \rangle.
\end{aligned}
$$

Similarly, we have

$$
\begin{aligned}
&\langle \mathbf{X}^{k+1} - \mathbf{X}^*, \mathbf{D}^{k+1} - \mathbf{D}^* \rangle \\
=&\langle (\mathbf{I} - \mathbf{W})(\mathbf{X}^{k+1} - \mathbf{X}^*), (\mathbf{I} - \mathbf{W})^\dagger (\mathbf{D}^{k+1} - \mathbf{D}^*) \rangle \\
=&\left\langle \frac{\eta}{\gamma}(2\mathbf{I} - \gamma(\mathbf{I} - \mathbf{W}))(\mathbf{D}^{k+1} - \mathbf{D}^k) - (\mathbf{I} - \mathbf{W})\mathbf{E}^k, (\mathbf{I} - \mathbf{W})^\dagger (\mathbf{D}^{k+1} - \mathbf{D}^*) \right\rangle \\
=&\left\langle \frac{\eta}{\gamma}(2(\mathbf{I} - \mathbf{W})^\dagger - \mathbf{I})(\mathbf{D}^{k+1} - \mathbf{D}^k) - \mathbf{E}^k, \mathbf{D}^{k+1} - \mathbf{D}^* \right\rangle \\
=&\frac{\eta}{\gamma}\langle \mathbf{D}^{k+1} - \mathbf{D}^k, \mathbf{D}^{k+1} - \mathbf{D}^* \rangle_{\mathbf{M}} - \langle \mathbf{E}^k, \mathbf{D}^{k+1} - \mathbf{D}^* \rangle.
\end{aligned}
$$

To make sure that $\mathbf{M}$ is positive definite over $\mathbf{span}\{\mathbf{I} - \mathbf{W}\}$, we need $\gamma < 2/\lambda_{\max}(\mathbf{I} - \mathbf{W})$. $\quad\square$

**Lemma 5.** *Taking the expectation conditioned on the compression in the $k$th iteration, we have*

$$
\begin{aligned}
2\eta\mathbb{E}\langle \mathbf{E}^k, \mathbf{D}^{k+1} - \mathbf{D}^* \rangle &= 2\eta\mathbb{E}\left\langle \mathbf{E}^k, \mathbf{D}^k + \frac{\gamma}{2\eta}(\mathbf{I} - \mathbf{W})\mathbf{Y}^k + \frac{\gamma}{2\eta}(\mathbf{I} - \mathbf{W})\mathbf{E}^k - \mathbf{D}^* \right\rangle \\
&= \gamma\mathbb{E}\langle \mathbf{E}^k, (\mathbf{I} - \mathbf{W})\mathbf{E}^k \rangle = \gamma\mathbb{E}\|\mathbf{E}^k\|_{\mathbf{I}-\mathbf{W}}^2, \\
2\eta\mathbb{E}\langle \mathbf{E}^k, \mathbf{D}^{k+1} - \mathbf{D}^k \rangle &= 2\eta\mathbb{E}\left\langle \mathbf{E}^k, \frac{\gamma}{2\eta}(\mathbf{I} - \mathbf{W})\mathbf{Y}^k + \frac{\gamma}{2\eta}(\mathbf{I} - \mathbf{W})\mathbf{E}^k \right\rangle \\
&= \gamma\mathbb{E}\langle \mathbf{E}^k, (\mathbf{I} - \mathbf{W})\mathbf{E}^k \rangle = \gamma\mathbb{E}\|\mathbf{E}^k\|_{\mathbf{I}-\mathbf{W}}^2.
\end{aligned}
$$

*Proof.* The proof is straightforward and omitted here. $\quad\square$

*Proof of Lemma 1.* From Alg. 1, we have

$$
\begin{aligned}
&2\eta\langle \mathbf{X}^k - \mathbf{X}^*, \nabla\mathbf{F}(\mathbf{X}^k; \xi^k) - \nabla\mathbf{F}(\mathbf{X}^*) \rangle \\
=&2\langle \mathbf{X}^k - \mathbf{X}^*, \eta\nabla\mathbf{F}(\mathbf{X}^k; \xi^k) - \eta\nabla\mathbf{F}(\mathbf{X}^*) \rangle \\
=&2\langle \mathbf{X}^k - \mathbf{X}^*, \mathbf{X}^k - \mathbf{X}^{k+1} - \eta(\mathbf{D}^{k+1} - \mathbf{D}^*) \rangle \quad \text{(from Line 7)} \\
=&2\langle \mathbf{X}^k - \mathbf{X}^*, \mathbf{X}^k - \mathbf{X}^{k+1} \rangle - 2\eta\langle \mathbf{X}^k - \mathbf{X}^*, \mathbf{D}^{k+1} - \mathbf{D}^* \rangle \\
=&2\langle \mathbf{X}^k - \mathbf{X}^*, \mathbf{X}^k - \mathbf{X}^{k+1} \rangle - 2\eta\langle \mathbf{X}^k - \mathbf{X}^{k+1}, \mathbf{D}^{k+1} - \mathbf{D}^* \rangle - 2\eta\langle \mathbf{X}^{k+1} - \mathbf{X}^*, \mathbf{D}^{k+1} - \mathbf{D}^* \rangle \\
=&2\langle \mathbf{X}^k - \mathbf{X}^* - \eta(\mathbf{D}^{k+1} - \mathbf{D}^*), \mathbf{X}^k - \mathbf{X}^{k+1} \rangle - 2\eta\langle \mathbf{X}^{k+1} - \mathbf{X}^*, \mathbf{D}^{k+1} - \mathbf{D}^* \rangle \\
=&2\langle \mathbf{X}^{k+1} - \mathbf{X}^* + \eta(\nabla\mathbf{F}(\mathbf{X}^k; \xi^k) - \nabla\mathbf{F}(\mathbf{X}^*)), \mathbf{X}^k - \mathbf{X}^{k+1} \rangle - 2\eta\langle \mathbf{X}^{k+1} - \mathbf{X}^*, \mathbf{D}^{k+1} - \mathbf{D}^* \rangle \quad \text{(from Line 7)} \\
=&2\langle \mathbf{X}^{k+1} - \mathbf{X}^*, \mathbf{X}^k - \mathbf{X}^{k+1} \rangle + 2\eta\langle \nabla\mathbf{F}(\mathbf{X}^k; \xi^k) - \nabla\mathbf{F}(\mathbf{X}^*), \mathbf{X}^k - \mathbf{X}^{k+1} \rangle \\
&- 2\eta\langle \mathbf{X}^{k+1} - \mathbf{X}^*, \mathbf{D}^{k+1} - \mathbf{D}^* \rangle.
\end{aligned} \tag{25}
$$

Then we consider the terms on the right hand side of (25) separately. Using $2\langle \mathbf{A} - \mathbf{B}, \mathbf{B} - \mathbf{C} \rangle = \|\mathbf{A} - \mathbf{C}\|^2 - \|\mathbf{B} - \mathbf{C}\|^2 - \|\mathbf{A} - \mathbf{B}\|^2$, we have

$$
\begin{aligned}
2\langle \mathbf{X}^{k+1} - \mathbf{X}^*, \mathbf{X}^k - \mathbf{X}^{k+1} \rangle =&2\langle \mathbf{X}^* - \mathbf{X}^{k+1}, \mathbf{X}^{k+1} - \mathbf{X}^k \rangle \\
=&\|\mathbf{X}^k - \mathbf{X}^*\|^2 - \|\mathbf{X}^{k+1} - \mathbf{X}^k\|^2 - \|\mathbf{X}^{k+1} - \mathbf{X}^*\|^2. \tag{26}
\end{aligned}
$$

Using $2\langle \mathbf{A}, \mathbf{B}\rangle = \|\mathbf{A}\|^2 + \|\mathbf{B}\|^2 - \|\mathbf{A} - \mathbf{B}\|^2$, we have

$$
\begin{aligned}
&2\eta\langle \nabla\mathbf{F}(\mathbf{X}^k;\xi^k) - \nabla\mathbf{F}(\mathbf{X}^*), \mathbf{X}^k - \mathbf{X}^{k+1}\rangle \\
=&\eta^2\|\nabla\mathbf{F}(\mathbf{X}^k;\xi^k) - \nabla\mathbf{F}(\mathbf{X}^*)\|^2 + \|\mathbf{X}^k - \mathbf{X}^{k+1}\|^2 - \|\mathbf{X}^k - \mathbf{X}^{k+1} - \eta(\nabla\mathbf{F}(\mathbf{X}^k;\xi^k) - \nabla\mathbf{F}(\mathbf{X}^*))\|^2 \\
=&\eta^2\|\nabla\mathbf{F}(\mathbf{X}^k;\xi^k) - \nabla\mathbf{F}(\mathbf{X}^*)\|^2 + \|\mathbf{X}^k - \mathbf{X}^{k+1}\|^2 - \eta^2\|\mathbf{D}^{k+1} - \mathbf{D}^*\|^2. \quad \text{(from Line 7)}
\end{aligned}
$$
(27)

Combining (25), (26), (27), and (23), we obtain

$$
\begin{aligned}
&2\eta\langle \mathbf{X}^k - \mathbf{X}^*, \nabla\mathbf{F}(\mathbf{X}^k;\xi^k) - \nabla\mathbf{F}(\mathbf{X}^*)\rangle \\
=& \underbrace{\|\mathbf{X}^k - \mathbf{X}^*\|^2 - \|\mathbf{X}^{k+1} - \mathbf{X}^k\|^2 - \|\mathbf{X}^{k+1} - \mathbf{X}^*\|^2}_{2\langle \mathbf{X}^{k+1} - \mathbf{X}^*, \mathbf{X}^k - \mathbf{X}^{k+1}\rangle} \\
&+ \underbrace{\eta^2\|\nabla\mathbf{F}(\mathbf{X}^k;\xi^k) - \nabla\mathbf{F}(\mathbf{X}^*)\|^2 + \|\mathbf{X}^k - \mathbf{X}^{k+1}\|^2 - \eta^2\|\mathbf{D}^{k+1} - \mathbf{D}^*\|^2}_{2\eta\langle \nabla\mathbf{F}(\mathbf{X}^k;\xi^k) - \nabla\mathbf{F}(\mathbf{X}^*), \mathbf{X}^k - \mathbf{X}^{k+1}\rangle} \\
&- \underbrace{\left( \frac{2\eta^2}{\gamma}\langle \mathbf{D}^{k+1} - \mathbf{D}^k, \mathbf{D}^{k+1} - \mathbf{D}^*\rangle_{\mathbf{M}} - 2\eta\langle \mathbf{E}^k, \mathbf{D}^{k+1} - \mathbf{D}^*\rangle \right)}_{2\eta\langle \mathbf{X}^{k+1} - \mathbf{X}^*, \mathbf{D}^{k+1} - \mathbf{D}^*\rangle} \\
=&\|\mathbf{X}^k - \mathbf{X}^*\|^2 - \|\mathbf{X}^{k+1} - \mathbf{X}^k\|^2 - \|\mathbf{X}^{k+1} - \mathbf{X}^*\|^2 \\
&+ \eta^2\|\nabla\mathbf{F}(\mathbf{X}^k;\xi^k) - \nabla\mathbf{F}(\mathbf{X}^*)\|^2 + \|\mathbf{X}^k - \mathbf{X}^{k+1}\|^2 - \eta^2\|\mathbf{D}^{k+1} - \mathbf{D}^*\|^2 \\
&+ \frac{\eta^2}{\gamma}\Big( \underbrace{\|\mathbf{D}^k - \mathbf{D}^*\|_{\mathbf{M}}^2 - \|\mathbf{D}^{k+1} - \mathbf{D}^*\|_{\mathbf{M}}^2 - \|\mathbf{D}^{k+1} - \mathbf{D}^k\|_{\mathbf{M}}^2}_{-2\langle \mathbf{D}^{k+1} - \mathbf{D}^k, \mathbf{D}^{k+1} - \mathbf{D}^*\rangle_{\mathbf{M}}}\Big) + 2\eta\langle \mathbf{E}^k, \mathbf{D}^{k+1} - \mathbf{D}^*\rangle,
\end{aligned}
$$

where the last equality holds because

$$
2\langle \mathbf{D}^k - \mathbf{D}^{k+1}, \mathbf{D}^{k+1} - \mathbf{D}^*\rangle_{\mathbf{M}} = \|\mathbf{D}^k - \mathbf{D}^*\|_{\mathbf{M}}^2 - \|\mathbf{D}^{k+1} - \mathbf{D}^*\|_{\mathbf{M}}^2 - \|\mathbf{D}^{k+1} - \mathbf{D}^k\|_{\mathbf{M}}^2.
$$

Thus, we reformulate it as

$$
\begin{aligned}
&\|\mathbf{X}^{k+1} - \mathbf{X}^*\|^2 + \frac{\eta^2}{\gamma}\|\mathbf{D}^{k+1} - \mathbf{D}^*\|_{\mathbf{M}}^2 \\
=&\|\mathbf{X}^k - \mathbf{X}^*\|^2 + \frac{\eta^2}{\gamma}\|\mathbf{D}^k - \mathbf{D}^*\|_{\mathbf{M}}^2 - \frac{\eta^2}{\gamma}\|\mathbf{D}^{k+1} - \mathbf{D}^k\|_{\mathbf{M}}^2 - \eta^2\|\mathbf{D}^{k+1} - \mathbf{D}^*\|^2 \\
&- 2\eta\langle \mathbf{X}^k - \mathbf{X}^*, \nabla\mathbf{F}(\mathbf{X}^k;\xi^k) - \nabla\mathbf{F}(\mathbf{X}^*)\rangle + \eta^2\|\nabla\mathbf{F}(\mathbf{X}^k;\xi^k) - \nabla\mathbf{F}(\mathbf{X}^*)\|^2 + 2\eta\langle \mathbf{E}^k, \mathbf{D}^{k+1} - \mathbf{D}^*\rangle,
\end{aligned}
$$

which completes the proof. $\qquad\square$

### E.4 PROOF OF LEMMA 2

*Proof of Lemma 2.* From Alg. 1, we take the expectation conditioned on $k$th compression and obtain

$$
\begin{aligned}
&\mathbb{E}\|\mathbf{H}^{k+1} - \mathbf{X}^*\|^2 \\
=&\mathbb{E}\|(1-\alpha)(\mathbf{H}^k - \mathbf{X}^*) + \alpha(\mathbf{Y}^k - \mathbf{X}^*) + \alpha\mathbf{E}^k\|^2 \quad \text{(from Line 13)} \\
=&\|(1-\alpha)(\mathbf{H}^k - \mathbf{X}^*) + \alpha(\mathbf{Y}^k - \mathbf{X}^*)\|^2 + \alpha^2\mathbb{E}\|\mathbf{E}^k\|^2 \\
=&(1-\alpha)\|\mathbf{H}^k - \mathbf{X}^*\|^2 + \alpha\|\mathbf{Y}^k - \mathbf{X}^*\|^2 - \alpha(1-\alpha)\|\mathbf{H}^k - \mathbf{Y}^k\|^2 + \alpha^2\mathbb{E}\|\mathbf{E}^k\|^2.
\end{aligned}
$$
(28)

In the second equality, we used the unbiasedness of the compression, i.e., $\mathbb{E}\mathbf{E}^k = \mathbf{0}$. The last equality holds because of

$$
\|(1-\alpha)\mathbf{A} + \alpha\mathbf{B}\|^2 = (1-\alpha)\|\mathbf{A}\|^2 + \alpha\|\mathbf{B}\|^2 - \alpha(1-\alpha)\|\mathbf{A} - \mathbf{B}\|^2.
$$

In addition, by taking the conditional expectation on the compression, we have

$$
\begin{aligned}
\|\mathbf{Y}^k - \mathbf{X}^*\|^2 =& \|\mathbf{X}^k - \eta \nabla \mathbf{F}(\mathbf{X}^k; \xi^k) - \eta \mathbf{D}^k - \mathbf{X}^*\|^2 \qquad \text{(from Line 4)} \\
=& \mathbb{E}\|\mathbf{X}^{k+1} + \eta \mathbf{D}^{k+1} - \eta \mathbf{D}^k - \mathbf{X}^*\|^2 \qquad \text{(from Line 7)} \\
=& \mathbb{E}\|\mathbf{X}^{k+1} - \mathbf{X}^*\|^2 + \eta^2 \mathbb{E}\|\mathbf{D}^{k+1} - \mathbf{D}^k\|^2 + 2\eta \mathbb{E}\langle \mathbf{X}^{k+1} - \mathbf{X}^*, \mathbf{D}^{k+1} - \mathbf{D}^k \rangle \\
=& \mathbb{E}\|\mathbf{X}^{k+1} - \mathbf{X}^*\|^2 + \eta^2 \mathbb{E}\|\mathbf{D}^{k+1} - \mathbf{D}^k\|^2 \\
& + \frac{2\eta^2}{\gamma} \mathbb{E}\|\mathbf{D}^{k+1} - \mathbf{D}^k\|_{\mathbf{M}}^2 - 2\eta \mathbb{E}\langle \mathbf{E}^k, \mathbf{D}^{k+1} - \mathbf{D}^k \rangle. \qquad \text{(from (23))} \\
=& \mathbb{E}\|\mathbf{X}^{k+1} - \mathbf{X}^*\|^2 + \eta^2 \mathbb{E}\|\mathbf{D}^{k+1} - \mathbf{D}^k\|^2 \\
& + \frac{2\eta^2}{\gamma} \mathbb{E}\|\mathbf{D}^{k+1} - \mathbf{D}^k\|_{\mathbf{M}}^2 - \gamma \mathbb{E}\|\mathbf{E}^k\|_{\mathbf{I}-\mathbf{W}}^2. \qquad \text{(from Line 6)} \qquad (29)
\end{aligned}
$$

Combing the above two equations (28) and (29) together, we have

$$
\mathbb{E}\|\mathbf{H}^{k+1} - \mathbf{X}^*\|^2
$$

$$
\begin{aligned}
\leq& (1-\alpha)\|\mathbf{H}^k - \mathbf{X}^*\|^2 + \alpha \mathbb{E}\|\mathbf{X}^{k+1} - \mathbf{X}^*\|^2 + \alpha \eta^2 \mathbb{E}\|\mathbf{D}^{k+1} - \mathbf{D}^k\|^2 + \frac{2\alpha \eta^2}{\gamma} \mathbb{E}\|\mathbf{D}^{k+1} - \mathbf{D}^k\|_{\mathbf{M}}^2 \\
& - \alpha \gamma \mathbb{E}\|\mathbf{E}^k\|_{\mathbf{I}-\mathbf{W}}^2 + \alpha^2 \mathbb{E}\|\mathbf{E}^k\|^2 - \alpha(1-\alpha)\|\mathbf{Y}^k - \mathbf{H}^k\|^2, \qquad (30)
\end{aligned}
$$

which completes the proof. $\qquad \square$

## E.5 PROOF OF THEOREM 1

*Proof of Theorem 1.* Combining Lemmas 1, 2, and 5, we have the expectation conditioned on the compression satisfying

$$
\mathbb{E}\|\mathbf{X}^{k+1} - \mathbf{X}^*\|^2 + \frac{\eta^2}{\gamma} \mathbb{E}\|\mathbf{D}^{k+1} - \mathbf{D}^*\|_{\mathbf{M}}^2 + a_1 \mathbb{E}\|\mathbf{H}^{k+1} - \mathbf{X}^*\|^2
$$

$$
\begin{aligned}
\leq& \|\mathbf{X}^k - \mathbf{X}^*\|^2 + \frac{\eta^2}{\gamma}\|\mathbf{D}^k - \mathbf{D}^*\|_{\mathbf{M}}^2 - \frac{\eta^2}{\gamma}\mathbb{E}\|\mathbf{D}^{k+1} - \mathbf{D}^k\|_{\mathbf{M}}^2 - \eta^2 \mathbb{E}\|\mathbf{D}^{k+1} - \mathbf{D}^*\|^2 \\
& - 2\eta\langle \mathbf{X}^k - \mathbf{X}^*, \nabla\mathbf{F}(\mathbf{X}^k;\xi^k) - \nabla\mathbf{F}(\mathbf{X}^*)\rangle + \eta^2\|\nabla\mathbf{F}(\mathbf{X}^k;\xi^k) - \nabla\mathbf{F}(\mathbf{X}^*)\|^2 + \gamma\mathbb{E}\|\mathbf{E}^k\|_{\mathbf{I}-\mathbf{W}}^2 \\
& + a_1(1-\alpha)\|\mathbf{H}^k - \mathbf{X}^*\|^2 + a_1\alpha\mathbb{E}\|\mathbf{X}^{k+1} - \mathbf{X}^*\|^2 + a_1\alpha\eta^2\mathbb{E}\|\mathbf{D}^{k+1} - \mathbf{D}^k\|^2 \\
& + \frac{2a_1\alpha\eta^2}{\gamma}\mathbb{E}\|\mathbf{D}^{k+1} - \mathbf{D}^k\|_{\mathbf{M}}^2 + a_1\alpha^2\mathbb{E}\|\mathbf{E}^k\|^2 - a_1\alpha\gamma\mathbb{E}\|\mathbf{E}^k\|_{\mathbf{I}-\mathbf{W}}^2 - a_1\alpha(1-\alpha)\|\mathbf{Y}^k - \mathbf{H}^k\|^2 \\
=& \underbrace{\|\mathbf{X}^k - \mathbf{X}^*\|^2 - 2\eta\langle \mathbf{X}^k - \mathbf{X}^*, \nabla\mathbf{F}(\mathbf{X}^k;\xi^k) - \nabla\mathbf{F}(\mathbf{X}^*)\rangle + \eta^2\|\nabla\mathbf{F}(\mathbf{X}^k;\xi^k) - \nabla\mathbf{F}(\mathbf{X}^*)\|^2}_{\mathcal{A}} \\
& + a_1\alpha\mathbb{E}\|\mathbf{X}^{k+1} - \mathbf{X}^*\|^2 + \frac{\eta^2}{\gamma}\|\mathbf{D}^k - \mathbf{D}^*\|_{\mathbf{M}}^2 - \eta^2\mathbb{E}\|\mathbf{D}^{k+1} - \mathbf{D}^*\|^2 \\
& + a_1(1-\alpha)\|\mathbf{H}^k - \mathbf{X}^*\|^2 \underbrace{-(1-2a_1\alpha)\frac{\eta^2}{\gamma}\mathbb{E}\|\mathbf{D}^{k+1} - \mathbf{D}^k\|_{\mathbf{M}}^2 + a_1\alpha\eta^2\mathbb{E}\|\mathbf{D}^{k+1} - \mathbf{D}^k\|^2}_{\mathcal{B}} \\
& + \underbrace{a_1\alpha^2\mathbb{E}\|\mathbf{E}^k\|^2 + (1-a_1\alpha)\gamma\mathbb{E}\|\mathbf{E}^k\|_{\mathbf{I}-\mathbf{W}}^2 - a_1\alpha(1-\alpha)\|\mathbf{Y}^k - \mathbf{H}^k\|^2}_{\mathcal{C}}, \qquad (31)
\end{aligned}
$$

where $a_1$ is a non-negative number to be determined. Then we deal with the three terms on the right hand side separately. We want the terms $\mathcal{B}$ and $\mathcal{C}$ to be nonpositive. First, we consider $\mathcal{B}$. Note that $\mathbf{D}^k \in \mathbf{Range}(\mathbf{I} - \mathbf{W})$. If we want $\mathcal{B} \leq 0$, then, we need $1 - 2a_1\alpha > 0$, i.e., $a_1\alpha < 1/2$. Therefore we have

$$
\begin{aligned}
\mathcal{B} =& -(1-2a_1\alpha)\frac{\eta^2}{\gamma}\mathbb{E}\|\mathbf{D}^{k+1} - \mathbf{D}^k\|_{\mathbf{M}}^2 + a_1\alpha\eta^2\mathbb{E}\|\mathbf{D}^{k+1} - \mathbf{D}^k\|^2 \\
\leq& \left(a_1\alpha - \frac{(1-2a_1\alpha)\lambda_{n-1}(\mathbf{M})}{\gamma}\right)\eta^2\mathbb{E}\|\mathbf{D}^{k+1} - \mathbf{D}^k\|^2,
\end{aligned}
$$

where $\lambda_{n-1}(\mathbf{M}) > 0$ is the second smallest eigenvalue of $\mathbf{M}$. It means that we also need

$$a_1\alpha + \frac{(2a_1\alpha - 1)\lambda_{n-1}(\mathbf{M})}{\gamma} \leq 0,$$

which is equivalent to

$$a_1\alpha \leq \frac{\lambda_{n-1}(\mathbf{M})}{\gamma + 2\lambda_{n-1}(\mathbf{M})} < 1/2. \tag{32}$$

Then we look at $\mathcal{C}$. We have

$$\begin{aligned}
\mathcal{C} &= a_1\alpha^2\mathbb{E}\|\mathbf{E}^k\|^2 + (1 - a_1\alpha)\gamma\mathbb{E}\|\mathbf{E}^k\|^2_{\mathbf{I}-\mathbf{W}} - a_1\alpha(1-\alpha)\|\mathbf{Y}^k - \mathbf{H}^k\|^2 \\
&\leq ((1 - a_1\alpha)\beta\gamma + a_1\alpha^2)\mathbb{E}\|\mathbf{E}^k\|^2 - a_1\alpha(1-\alpha)\|\mathbf{Y}^k - \mathbf{H}^k\|^2 \\
&\leq C((1 - a_1\alpha)\beta\gamma + a_1\alpha^2)\|\mathbf{Y}^k - \mathbf{H}^k\|^2 - a_1\alpha(1-\alpha)\|\mathbf{Y}^k - \mathbf{H}^k\|^2
\end{aligned}$$

Because we have $1 - a_1\alpha > 1/2$, so we need

$$C((1 - a_1\alpha)\beta\gamma + a_1\alpha^2) - a_1\alpha(1-\alpha) = (1+C)a_1\alpha^2 - a_1(C\beta\gamma + 1)\alpha + C\beta\gamma \leq 0. \tag{33}$$

That is

$$\alpha \geq \frac{a_1(C\beta\gamma + 1) - \sqrt{a_1^2(C\beta\gamma + 1)^2 - 4(1+C)Ca_1\beta\gamma}}{2(1+C)a_1} =: \alpha_0, \tag{34}$$

$$\alpha \leq \frac{a_1(C\beta\gamma + 1) + \sqrt{a_1^2(C\beta\gamma + 1)^2 - 4(1+C)Ca_1\beta\gamma}}{2(1+C)a_1} =: \alpha_1. \tag{35}$$

Next, we look at $\mathcal{A}$. Firstly, by the bounded variance assumption, we have the expectation conditioned on the gradient sampling in $k$th iteration satisfying

$$\begin{aligned}
&\mathbb{E}\|\mathbf{X}^k - \mathbf{X}^*\|^2 - 2\eta\mathbb{E}\langle\mathbf{X}^k - \mathbf{X}^*, \nabla\mathbf{F}(\mathbf{X}^k; \xi^k) - \nabla\mathbf{F}(\mathbf{X}^*)\rangle + \eta^2\mathbb{E}\|\nabla\mathbf{F}(\mathbf{X}^k; \xi^k) - \nabla\mathbf{F}(\mathbf{X}^*)\|^2 \\
&\leq \|\mathbf{X}^k - \mathbf{X}^*\|^2 - 2\eta\langle\mathbf{X}^k - \mathbf{X}^*, \nabla\mathbf{F}(\mathbf{X}^k) - \nabla\mathbf{F}(\mathbf{X}^*)\rangle + \eta^2\|\nabla\mathbf{F}(\mathbf{X}^k) - \nabla\mathbf{F}(\mathbf{X}^*)\|^2 + n\eta^2\sigma^2
\end{aligned}$$

Then with the smoothness and strong convexity from Assumptions 4, we have the co-coercivity of $\nabla g_i(\boldsymbol{x})$ with $g_i(\boldsymbol{x}) := f_i(\boldsymbol{x}) - \frac{u}{2}\|\boldsymbol{x}\|_2^2$, which gives

$$\langle\mathbf{X}^k - \mathbf{X}^*, \nabla\mathbf{F}(\mathbf{X}^k) - \nabla\mathbf{F}(\mathbf{X}^*)\rangle \geq \frac{\mu L}{\mu + L}\|\mathbf{X}^k - \mathbf{X}^*\|^2 + \frac{1}{\mu + L}\|\nabla\mathbf{F}(\mathbf{X}^k) - \nabla\mathbf{F}(\mathbf{X}^*)\|^2.$$

When $\eta \leq 2/(\mu + L)$, we have

$$\begin{aligned}
&\langle\mathbf{X}^k - \mathbf{X}^*, \nabla\mathbf{F}(\mathbf{X}^k) - \nabla\mathbf{F}(\mathbf{X}^*)\rangle \\
&= \left(1 - \frac{\eta(\mu + L)}{2}\right)\langle\mathbf{X}^k - \mathbf{X}^*, \nabla\mathbf{F}(\mathbf{X}^k) - \nabla\mathbf{F}(\mathbf{X}^*)\rangle + \frac{\eta(\mu + L)}{2}\langle\mathbf{X}^k - \mathbf{X}^*, \nabla\mathbf{F}(\mathbf{X}^k) - \nabla\mathbf{F}(\mathbf{X}^*)\rangle \\
&\geq \left(\mu - \frac{\eta\mu(\mu + L)}{2} + \frac{\eta\mu L}{2}\right)\|\mathbf{X}^k - \mathbf{X}^*\|^2 + \frac{\eta}{2}\|\nabla\mathbf{F}(\mathbf{X}^k) - \nabla\mathbf{F}(\mathbf{X}^*)\|^2 \\
&= \mu\left(1 - \frac{\eta\mu}{2}\right)\|\mathbf{X}^k - \mathbf{X}^*\|^2 + \frac{\eta}{2}\|\nabla\mathbf{F}(\mathbf{X}^k) - \nabla\mathbf{F}(\mathbf{X}^*)\|^2.
\end{aligned}$$

Therefore, we obtain

$$\begin{aligned}
&-2\eta\langle\mathbf{X}^k - \mathbf{X}^*, \nabla\mathbf{F}(\mathbf{X}^k) - \nabla\mathbf{F}(\mathbf{X}^*)\rangle \\
&\leq -\eta^2\|\nabla\mathbf{F}(\mathbf{X}^k) - \nabla\mathbf{F}(\mathbf{X}^*)\|^2 - \mu(2\eta - \mu\eta^2)\|\mathbf{X}^k - \mathbf{X}^*\|^2. \tag{36}
\end{aligned}$$

Conditioned on the $k$the iteration, (i.e., conditioned on the gradient sampling in $k$th iteration), the inequality (31) becomes

$$\begin{aligned}
&\mathbb{E}\|\mathbf{X}^{k+1} - \mathbf{X}^*\|^2 + \frac{\eta^2}{\gamma}\mathbb{E}\|\mathbf{D}^{k+1} - \mathbf{D}^*\|^2_{\mathbf{M}} + a_1\mathbb{E}\|\mathbf{H}^{k+1} - \mathbf{X}^*\|^2 \\
&\leq \left(1 - \mu(2\eta - \mu\eta^2)\right)\|\mathbf{X}^k - \mathbf{X}^*\|^2 + a_1\alpha\mathbb{E}\|\mathbf{X}^{k+1} - \mathbf{X}^*\|^2 \\
&\quad + \frac{\eta^2}{\gamma}\|\mathbf{D}^k - \mathbf{D}^*\|^2_{\mathbf{M}} - \eta^2\mathbb{E}\|\mathbf{D}^{k+1} - \mathbf{D}^*\|^2 + a_1(1-\alpha)\|\mathbf{H}^k - \mathbf{X}^*\|^2 + n\eta^2\sigma^2, \tag{37}
\end{aligned}$$

if the step size satisfies $\eta \leq \frac{2}{\mu+L}$. Rewriting (37), we have

$$(1 - a_1\alpha)\mathbb{E}\|\mathbf{X}^{k+1} - \mathbf{X}^*\|^2 + \frac{\eta^2}{\gamma}\mathbb{E}\|\mathbf{D}^{k+1} - \mathbf{D}^*\|_\mathbf{M}^2 + \eta^2\mathbb{E}\|\mathbf{D}^{k+1} - \mathbf{D}^*\|^2 + a_1\mathbb{E}\|\mathbf{H}^{k+1} - \mathbf{X}^*\|^2$$

$$\leq \left(1 - \mu(2\eta - \mu\eta^2)\right)\|\mathbf{X}^k - \mathbf{X}^*\|^2 + \frac{\eta^2}{\gamma}\|\mathbf{D}^k - \mathbf{D}^*\|_\mathbf{M}^2 + a_1(1-\alpha)\|\mathbf{H}^k - \mathbf{X}^*\|^2 + n\eta^2\sigma^2, \tag{38}$$

and thus

$$(1 - a_1\alpha)\mathbb{E}\|\mathbf{X}^{k+1} - \mathbf{X}^*\|^2 + \frac{\eta^2}{\gamma}\mathbb{E}\|\mathbf{D}^{k+1} - \mathbf{D}^*\|_{\mathbf{M}+\gamma\mathbf{I}}^2 + a_1\mathbb{E}\|\mathbf{H}^{k+1} - \mathbf{X}^*\|^2$$

$$\leq \left(1 - \mu(2\eta - \mu\eta^2)\right)\|\mathbf{X}^k - \mathbf{X}^*\|^2 + \frac{\eta^2}{\gamma}\|\mathbf{D}^k - \mathbf{D}^*\|_\mathbf{M}^2 + a_1(1-\alpha)\|\mathbf{H}^k - \mathbf{X}^*\|^2 + n\eta^2\sigma^2. \tag{39}$$

With the definition of $\mathcal{L}^k$ in (12), we have

$$\mathbb{E}\mathcal{L}^{k+1} \leq \rho\mathcal{L}^k + n\eta^2\sigma^2, \tag{40}$$

with

$$\rho = \max\left\{\frac{1 - \mu(2\eta - \mu\eta^2)}{1 - a_1\alpha}, \frac{\lambda_{\max}(\mathbf{M})}{\gamma + \lambda_{\max}(\mathbf{M})}, 1 - \alpha\right\}.$$

where

$$\lambda_{\max}(\mathbf{M}) = 2\lambda_{\max}((\mathbf{I} - \mathbf{W})^\dagger) - \gamma.$$

Recall all the conditions on the parameters $a_1$, $\alpha$, and $\gamma$ to make sure that $\rho < 1$:

$$a_1\alpha \leq \frac{\lambda_{n-1}(\mathbf{M})}{\gamma + 2\lambda_{n-1}(\mathbf{M})}, \tag{41}$$

$$a_1\alpha \leq \mu(2\eta - \mu\eta^2), \tag{42}$$

$$\alpha \geq \frac{a_1(C\beta\gamma + 1) - \sqrt{a_1^2(C\beta\gamma + 1)^2 - 4(1 + C)Ca_1\beta\gamma}}{2(1 + C)a_1} =: \alpha_0, \tag{43}$$

$$\alpha \leq \frac{a_1(C\beta\gamma + 1) + \sqrt{a_1^2(C\beta\gamma + 1)^2 - 4(1 + C)Ca_1\beta\gamma}}{2(1 + C)a_1} =: \alpha_1. \tag{44}$$

In the following, we show that there exist parameters that satisfy these conditions.

Since we can choose any $a_1$, we let

$$a_1 = \frac{4(1 + C)}{C\beta\gamma + 2},$$

such that

$$a_1^2(C\beta\gamma + 1)^2 - 4(1 + C)Ca_1\beta\gamma = a_1^2.$$

Then we have

$$\alpha_0 = \frac{C\beta\gamma}{2(1 + C)} \to 0, \qquad \text{as } \gamma \to 0,$$

$$\alpha_1 = \frac{C\beta\gamma + 2}{2(1 + C)} \to \frac{1}{1 + C}, \quad \text{as } \gamma \to 0.$$

Conditions (43) and (44) show

$$a_1\alpha \in \left[\frac{2C\beta\gamma}{C\beta\gamma + 2}, 2\right] \to [0, 2], \text{ if } C = 0 \text{ or } \gamma \to 0.$$

Hence in order to make (41) and (42) satisfied, it's sufficient to make

$$\frac{2C\beta\gamma}{C\beta\gamma + 2} \leq \min\left\{\frac{\lambda_{n-1}(\mathbf{M})}{\gamma + 2\lambda_{n-1}(\mathbf{M})}, \mu(2\eta - \mu\eta^2)\right\} = \min\left\{\frac{\frac{2}{\beta} - \gamma}{\frac{4}{\beta} - \gamma}, \mu(2\eta - \mu\eta^2)\right\}. \quad (45)$$

where we use $\lambda_{n-1}(\mathbf{M}) = \frac{2}{\lambda_{\max}(\mathbf{I-W})} - \gamma = \frac{2}{\beta} - \gamma$.

When $C > 0$, the condition (45) is equivalent to

$$\gamma \leq \min\left\{\frac{(3C+1) - \sqrt{(3C+1)^2 - 4C}}{C\beta}, \frac{2\mu\eta(2-\mu\eta)}{[2 - \mu\eta(2-\mu\eta)]C\beta}\right\}. \quad (46)$$

The first term can be simplified using

$$\frac{(3C+1) - \sqrt{(3C+1)^2 - 4C}}{C\beta} \geq \frac{2}{(3C+1)\beta}$$

due to $\sqrt{1-x} \leq 1 - \frac{x}{2}$ when $x \in (0,1)$.

Therefore, for a given stepsize $\eta$, if we choose

$$\gamma \in \left(0, \min\left\{\frac{2}{(3C+1)\beta}, \frac{2\mu\eta(2-\mu\eta)}{[2 - \mu\eta(2-\mu\eta)]C\beta}\right\}\right)$$

and

$$\alpha \in \left[\frac{C\beta\gamma}{2(1+C)}, \min\left\{\frac{C\beta\gamma + 2}{2(1+C)}, \frac{2-\beta\gamma}{4-\beta\gamma}\frac{C\beta\gamma + 2}{4(1+C)}, \mu\eta(2-\mu\eta)\frac{C\beta\gamma + 2}{4(1+C)}\right\}\right],$$

then, all conditions (41)-(44) hold.

Note that $\gamma < \frac{2}{(3C+1)\beta}$ implies $\gamma < \frac{2}{\beta}$, which ensures the positive definiteness of $\mathbf{M}$ over $\mathbf{span}\{\mathbf{I} - \mathbf{W}\}$ in Lemma 4.

Note that $\eta \leq \frac{2}{\mu + L}$ ensures

$$\mu\eta(2-\mu\eta)\frac{C\beta\gamma + 2}{4(1+C)} \leq \frac{C\beta\gamma + 2}{2(1+C)}. \quad (47)$$

So, we can simplify the bound for $\alpha$ as

$$\alpha \in \left[\frac{C\beta\gamma}{2(1+C)}, \min\left\{\frac{2-\beta\gamma}{4-\beta\gamma}\frac{C\beta\gamma + 2}{4(1+C)}, \mu\eta(2-\mu\eta)\frac{C\beta\gamma + 2}{4(1+C)}\right\}\right].$$

Lastly, taking the total expectation on both sides of (40) and using tower property, we complete the proof for $C > 0$.

$$\square$$

*Proof of Corollary 1.* Let's first define $\kappa_f = \frac{L}{\mu}$ and $\kappa_g = \frac{\lambda_{\max}(\mathbf{I-W})}{\lambda_{\min}^+(\mathbf{I-W})} = \lambda_{\max}(\mathbf{I} - \mathbf{W})\lambda_{\max}((\mathbf{I} - \mathbf{W})^\dagger)$.

We can choose the stepsize $\eta = \frac{1}{L}$ such that the upper bound of $\gamma$ is

$$\gamma_{\text{upper}} = \min\left\{\frac{2}{(3C+1)\beta}, \frac{\frac{2}{\kappa_f}\left(2 - \frac{1}{\kappa_f}\right)}{\left[2 - \frac{1}{\kappa_f}\left(2 - \frac{1}{\kappa_f}\right)\right]C\beta}, \frac{2}{\beta}\right\} \geq \min\left\{\frac{2}{(3C+1)\beta}, \frac{1}{\kappa_f C\beta}\right\},$$

due to $\frac{x(2-x)}{2-x(2-x)} \geq \frac{x}{2-x} \geq x$ when $x \in (0,1)$.

Hence we can take $\gamma = \min\{\frac{1}{(3C+1)\beta}, \frac{1}{\kappa_f C\beta}\}$.

The bound of $\alpha$ is

$$\alpha \in \left[ \frac{C\beta\gamma}{2(1+C)}, \min\left\{ \frac{2-\beta\gamma}{4-\beta\gamma}\frac{C\beta\gamma+2}{4(1+C)}, \frac{1}{\kappa_f}(2-\frac{1}{\kappa_f})\frac{C\beta\gamma+2}{4(1+C)} \right\} \right]$$

When $\gamma$ is chosen as $\frac{1}{\kappa_f C\beta}$, pick

$$\alpha = \frac{C\beta\gamma}{2(1+C)} = \frac{1}{2(1+C)\kappa_f}. \tag{48}$$

When $\frac{1}{(3C+1)\beta} \leq \frac{1}{\kappa_f C\beta}$, the upper bound of $\alpha$ is

$$\begin{aligned}
\alpha_{\text{upper}} &= \min\left\{ \frac{2-\beta\gamma}{4-\beta\gamma}\frac{C\beta\gamma+2}{4(1+C)}, \frac{1}{\kappa_f}(2-\frac{1}{\kappa_f})\frac{C\beta\gamma+2}{4(1+C)} \right\} \\
&= \min\left\{ \frac{6C+1}{12C+3}, \frac{1}{\kappa_f}(2-\frac{1}{\kappa_f}) \right\} \frac{7C+2}{4(C+1)(3C+1)} \\
&\geq \min\left\{ \frac{6C+1}{12C+3}, \frac{1}{\kappa_f} \right\} \frac{7C+2}{4(C+1)(3C+1)}.
\end{aligned}$$

In this case, we pick

$$\alpha = \min\left\{ \frac{6C+1}{12C+3}, \frac{1}{\kappa_f} \right\} \frac{7C+2}{4(C+1)(3C+1)}. \tag{49}$$

Note $\alpha = \mathcal{O}\left( \frac{1}{(1+C)\kappa_f} \right)$ since $\frac{6C+1}{12C+3}$ is lower bounded by $\frac{1}{3}$. Hence in both cases (Eq. (48) and Eq. (49)), $\alpha = \mathcal{O}\left( \frac{1}{(1+C)\kappa_f} \right)$, and the third term of $\rho$ is upper bounded by

$$1 - \alpha \leq \max\left\{ 1 - \frac{1}{2(1+C)\kappa_f}, 1 - \min\left\{ \frac{6C+1}{12C+3}, \frac{1}{\kappa_f} \right\} \frac{7C+2}{4(1+C)(3C+1)} \right\}$$

In two cases of $\gamma$, the second term of $\rho$ becomes

$$1 - \frac{\gamma}{2\lambda_{\max}((\mathbf{I}-\mathbf{W})^\dagger)} = \max\left\{ 1 - \frac{1}{2C\kappa_f\kappa_g}, 1 - \frac{1}{(1+3C)\kappa_g} \right\}$$

Before analysing the first term of $\rho$, we look at $a_1\alpha$ in two cases of $\gamma$. When $\gamma = \frac{1}{\kappa_f C\beta}$,

$$a_1\alpha = \frac{2C\beta\gamma}{C\beta\gamma+2} = \frac{2}{2\kappa_f+1} \leq \frac{1}{\kappa_f}.$$

When $\gamma = \frac{1}{(3C+1)\beta}$,

$$a_1\alpha = \min\left\{ \frac{6C+1}{(12C+3)}, \frac{1}{\kappa_f} \right\} \leq \frac{1}{\kappa_f}.$$

In both cases, $a_1\alpha \leq \frac{1}{\kappa_f}$. Therefore, the first term of $\rho$ becomes

$$\frac{1-\mu\eta(2-\mu\eta)}{1-a_1\alpha} \leq \frac{1-\frac{1}{\kappa_f}(2-\frac{1}{\kappa_f})}{1-\frac{1}{\kappa_f}} = 1 - \frac{1-\frac{1}{\kappa_f}}{\kappa_f-1} = 1 - \frac{1}{\kappa_f}.$$

To summarize, we have

$$\rho \leq 1 - \min\left\{ \frac{1}{\kappa_f}, \frac{1}{2C\kappa_f\kappa_g}, \frac{1}{(1+3C)\kappa_g}, \frac{1}{2(1+C)\kappa_f}, \min\left\{ \frac{6C+1}{12C+3}, \frac{1}{\kappa_f} \right\} \frac{7C+2}{4(1+C)(3C+1)} \right\}$$

and therefore

$$\rho = \max\left\{1 - \mathcal{O}\left(\frac{1}{(1+C)\kappa_f}\right), 1 - \mathcal{O}\left(\frac{1}{(1+C)\kappa_g}\right), 1 - \mathcal{O}\left(\frac{1}{C\kappa_f\kappa_g}\right)\right\}.$$

With full-gradient (i.e., $\sigma = 0$), we get $\epsilon-$accuracy solution with the total number of iterations

$$k \geq \widetilde{\mathcal{O}}((1+C)(\kappa_f + \kappa_g) + C\kappa_f\kappa_g).$$

When $C = 0$, i.e., there is no compression, the iteration complexity recovers that of NIDS, $\widetilde{\mathcal{O}}\left(\kappa_f + \kappa_g\right)$.

When $C \leq \frac{\kappa_f + \kappa_g}{\kappa_f\kappa_g + \kappa_f + \kappa_g}$, the complexity is improved to that of NIDS, i.e., the compression doesn't harm the convergence in terms of the order of the coefficients. □

*Proof of Corollary 2.* Note that $(\bar{\boldsymbol{x}}^k)^\top = \overline{\mathbf{X}}^k$ and $\mathbf{1}_{n\times1}\overline{\mathbf{X}}^* = \mathbf{X}^*$, then

$$\sum_{i=1}^n \mathbb{E}\|\boldsymbol{x}_i^k - \bar{\boldsymbol{x}}^k\|^2 = \mathbb{E}\left\|\mathbf{X}^k - \mathbf{1}_{n\times1}\overline{\mathbf{X}}^k\right\|^2$$

$$= \mathbb{E}\left\|\mathbf{X}^k - \mathbf{X}^* + \mathbf{X}^* - \mathbf{1}_{n\times1}\overline{\mathbf{X}}^k\right\|^2$$

$$= \mathbb{E}\left\|\mathbf{X}^k - \mathbf{X}^* - \frac{\mathbf{1}_{n\times1}\mathbf{1}_{n\times1}^\top}{n}\left(\mathbf{X}^k - \mathbf{X}^*\right)\right\|$$

$$\leq \mathbb{E}\|\mathbf{X}^k - \mathbf{X}^*\|^2$$

$$\leq \frac{\rho\mathbb{E}\mathcal{L}^{k-1} + n\eta^2\sigma^2(1-\rho)^{-1}}{1 - a_1\alpha}$$

$$\leq 2\rho^k\mathcal{L}^0 + 2\frac{n\eta^2\sigma^2}{1-\rho}. \tag{50}$$

The last inequality holds because we have $a_1\alpha \leq 1/2$. □

*Proof of Corollary 3.* From the proof of Theorem 1, when $C = 0$, we can set $\gamma = 1$, $\alpha = 1$, and $a_1 = 0$. Plug those values into $\rho$, and we obtain the convergence rate for NIDS. □

### E.6   PROOF OF THEOREM 2

*Proof of Theorem 2.* In order to get exact convergence, we pick diminishing step-size, set $\alpha = \frac{C\beta\gamma}{2(1+C)}$, $a_1\alpha = \frac{2C\beta\gamma_k}{C\beta\gamma_k+2}$, $\theta_1 = \frac{1}{2\lambda_{\max}((\mathbf{I}-\mathbf{W})^\dagger)}$ and $\theta_2 = \frac{C\beta}{2(1+C)}$, then

$$\rho_k = \max\left\{1 - \frac{\mu\eta_k(2-\mu\eta_k) - a_1\alpha}{1 - a_1\alpha}, 1 - \theta_1\gamma_k, 1 - \theta_2\gamma_k\right\}$$

If we further pick diminishing $\eta_k$ and $\gamma_k$ such that $\mu\eta_k(2-\mu\eta_k) - a_1\alpha \geq a_1\alpha$, then

$$\frac{\mu\eta_k(2-\mu\eta_k) - a_1\alpha}{1 - a_1\alpha} \geq \frac{a_1\alpha}{1 - a_1\alpha} = \frac{2C\beta\gamma_k}{2 - C\beta\gamma_k} \geq C\beta\gamma_k.$$

Notice that $C\beta\gamma \leq \frac{2}{3}$ since $(3C+1) - \sqrt{(3C+1)^2 - 4C}$ is increasing in $C > 0$ with limit $\frac{2}{3}$ at $\infty$.

In this case we only need,

$$\gamma_k \in \left(0, \min\left\{\frac{(3C+1) - \sqrt{(3C+1)^2 - 4C}}{C\beta}, \frac{2\mu\eta_k(2-\mu\eta_k)}{[4 - \mu\eta_k(2-\mu\eta_k)]C\beta}, \frac{2}{\beta}\right\}\right). \tag{51}$$

And

$$\rho_k \leq \max\left\{1 - C\beta\gamma_k, 1 - \theta_1\gamma_k, 1 - \theta_2\gamma_k\right\} \leq 1 - \theta_3\gamma_k$$

if $\theta_3 = \min\{\theta_1, \theta_2\}$ and note that $\theta_2 \leq C\beta$.

We define

$$\mathcal{L}^k := (1 - a_1\alpha_k)\|\mathbf{X}^k - \mathbf{X}^*\|^2 + (2\eta_k^2/\gamma_k)\mathbb{E}\|\mathbf{D}^{k+1} - \mathbf{D}^*\|^2_{(\mathbf{I}-\mathbf{W})^\dagger} + a_1\|\mathbf{H}^k - \mathbf{X}^*\|^2.$$

Hence

$$\mathbb{E}\mathcal{L}^{k+1} \leq (1 - \theta_3\gamma_k)\mathbb{E}\mathcal{L}^k + n\sigma^2\eta_k^2.$$

From $a_1\alpha \leq \frac{\mu\eta_k(2-\mu\eta_k)}{2}$, we get

$$\frac{4C\beta\gamma_k}{C\beta\gamma_k + 2} \leq \mu\eta_k(2 - \mu\eta_k).$$

If we pick $\gamma_k = \theta_4\eta_k$, then it's sufficient to let

$$2C\beta\theta_4\eta_k \leq \mu\eta_k(2 - \mu\eta_k).$$

Hence if $\theta_4 < \frac{\mu}{C\beta}$ and let $\eta_* = \frac{2(\mu-C\beta\theta_4)}{\mu^2}$, then $\eta_k = \frac{\gamma_k}{\theta_4} \in (0, \eta_*)$ guarantees the above discussion and

$$\mathbb{E}\mathcal{L}^{k+1} \leq (1 - \theta_3\theta_4\eta_k)\mathbb{E}\mathcal{L}^k + n\sigma^2\eta_k^2.$$

So far all restrictions for $\eta_k$ are

$$\eta_k \leq \min\left\{\frac{2}{\mu + L}, \eta_*\right\}$$

and

$$\eta_k \leq \frac{1}{\theta_4}\min\left\{\frac{(3C + 1) - \sqrt{(3C + 1)^2 - 4C}}{C\beta}, \frac{2}{\beta}\right\}.$$

Let $\theta_5 = \min\left\{\frac{2}{\mu+L}, \eta_*, \frac{(3C+1)-\sqrt{(3C+1)^2-4C}}{C\beta\theta_4}, \frac{2}{\beta\theta_4}\right\}$, $\eta_k = \frac{1}{Bk+A}$ and $D = \max\left\{A\mathcal{L}^0, \frac{2n\sigma^2}{\theta_3\theta_4}\right\}$, we claim that if we pick $B = \frac{\theta_3\theta_4}{2}$ and some $A$, by setting $\eta_k = \frac{2}{\theta_3\theta_4 k + 2A}$, we get

$$\mathbb{E}\mathcal{L}^k \leq \frac{D}{Bk + A}.$$

Induction:
When $k = 0$, it's obvious. Suppose previous $k$ inequalities hold. Then

$$\mathbb{E}\mathcal{L}^{k+1} \leq \left(1 - \frac{2\theta_3\theta_4}{\theta_3\theta_4 k + 2A}\right)\frac{2D}{\theta_3\theta_4 k + 2A} + \frac{4n\sigma^2}{(\theta_3\theta_4 k + 2A)^2}.$$

Multiply $M := (\theta_3\theta_4 k + \theta_3\theta_4 + 2A)(\theta_3\theta_4 k + 2A)(2D)^{-1}$ on both sides, we get

$$M\mathbb{E}\mathcal{L}^{k+1} \leq \left(1 - \frac{2\theta_3\theta_4}{\theta_3\theta_4 k + 2A}\right)(\theta_3\theta_4 k + \theta_3\theta_4 + 2A) + \frac{4n\sigma^2(\theta_3\theta_4 k + \theta_3\theta_4 + 2A)}{2D(\theta_3\theta_4 k + 2A)}$$

$$= \frac{2D(\theta_3\theta_4 k + 2A - 2\theta_3\theta_4)(\theta_3\theta_4 k + \theta_3\theta_4 + 2A) + 4n\sigma^2(\theta_3\theta_4 k + \theta_3\theta_4 + 2A)}{2D(\theta_3\theta_4 k + 2A)}$$

$$= \frac{2D(\theta_3\theta_4 k + 2A)^2 + 4n\sigma^2(\theta_3\theta_4 k + 2A) - 4D\theta_3\theta_4(\theta_3\theta_4 k + 2A) + 2D\theta_3\theta_4(\theta_3\theta_4 k + 2A)}{2D(\theta_3\theta_4 k + 2A)}$$

$$+ \frac{-4D(\theta_3\theta_4)^2 + 4n\sigma^2\theta_3\theta_4}{2D(\theta_3\theta_4 k + 2A)}$$

$$\leq \theta_3\theta_4 k + 2A.$$

Hence

$$\mathbb{E}\mathcal{L}^{k+1} \leq \frac{2D}{\theta_3\theta_4(k + 1) + 2A}$$

This induction holds for any $A$ such that $\eta_k$ is feasible, i.e.

$$\eta_0 = \frac{1}{A} \leq \theta_5.$$

Here we summarize the definition of constant numbers:

$$\theta_1 = \frac{1}{2\lambda_{\max}((\mathbf{I} - \mathbf{W})^\dagger)}, \; \theta_2 = \frac{C\beta}{2(1 + C)}, \tag{52}$$

$$\theta_3 = \min\{\theta_1, \theta_2\}, \; \theta_4 \in \left(0, \frac{\mu}{C\beta}\right), \; \eta_* = \frac{2(\mu - C\beta\theta_4)}{\mu^2}, \tag{53}$$

$$\theta_5 = \min\left\{\frac{2}{\mu + L}, \eta_*, \frac{(3C + 1) - \sqrt{(3C + 1)^2 - 4C}}{C\beta\theta_4}, \frac{2}{\beta\theta_4}\right\}. \tag{54}$$

Therefore, let $A = \frac{1}{\theta_5}$ and $\eta_k = \frac{2\theta_5}{\theta_3\theta_4\theta_5 k + 2}$, we get

$$\frac{1}{n}\mathbb{E}\mathcal{L}^k \leq \frac{2 \max\left\{\frac{1}{n}\mathcal{L}^0, \frac{2\sigma^2\theta_5}{\theta_3\theta_4}\right\}}{\theta_3\theta_4\theta_5 k + 2}.$$

Since $1 - a_1\alpha_k \geq 1/2$, we complete the proof.

$\square$

