# OpenReview forum: "Linear Convergent Decentralized Optimization with Compression"
_ICLR.cc/2021/Conference — ICLR 2021 Poster_

### Official Review · AnonReviewer1 · 2020-10-27
**Very nice work, with clear motivation and a good balance of theory and experiments**

**Rating:** 7
**Confidence:** 4

**Review:**

## Summary
This work proposes a new stochastic algorithm for distributed optimization. I really enjoyed reading the paper. It has a strong theory, nice experiments with several objectives and parameter-sensitivity tests, and the writing is sufficiently good. This authors did a great job comparing both theoretically and numerically to the most relevant literature. The paper builds on top of the recent successes in distributed optimization and two particularly popular approaches: quantization and decentralized topology of the network. The paper's main weakness, in my opinion, is that the algorithm is a bit hard to understand at first, and it has a number of extra parameters, but this seems to be a common issue for decentralized algorithms, and the experiments show that their tuning is not difficult. Another issue is that the paper might be too technical for an unprepared reader, and the theorem statement is hard to parse.

## Detailed comments
1. The paper presents a clear motivation for developing a new algorithm by explaining the shortcomings of the existing ones. For instance, D-PSGD is explained to not converge precisely when the data is heterogeneous; error-feedback has a slow rate due to the delayed compensation of errors; DCD-PSGD is mentioned in Remark 1 to be too aggressive and to perform worse numerically. I think that the thorough comparison to the prior work is one of the main strengths of this paper.

2. I think the algorithm is not explained well. Its design consists of multiple pieces: a consensus algorithm based on NIDS/D^2, a compression scheme based on Diana, and the gradient updates of SGD, all of which are given together immediately. The authors present the algorithm without any warm-up, and there are several ways to fix it. Firstly, the meaning of the variables that are presented in the algorithm could help to understand it. Before presenting the full algorithm, it makes sense to mention that D^k is a dual variable needed to ensure stationarity at the optimum and that its role is to estimate the gradient there. Similarly, the communication procedure is not obvious and the meaning of H^k is unclear when just looking at the algorithmic steps. I do think that the paper would benefit from introducing some of the concepts before the algorithm. My first impression was a lot of confusion, mainly because of the communication procedure. Maybe it's best if the authors explain what happens when the algorithm is fully centralized, what it boils down to, and why it works in that case.

3. Theorem 1 assumes that the gradients are bounded uniformly over the space. A number of recent papers showed that SGD works even if the variance is bounded only at the optimum, for example, (Gower et al., "SGD: General Analysis and Improved Rates"). Do the authors think that it's possible to relax the assumptions for LEAD as well?

4. Can the authors present a complexity bound based on Theorem 1 that would show explicitly every term? I think it should be something O((C*kappa+beta)*log(1/eps) + kappa*sigma^2/eps) and I'd hope to see a comparison to the bound of (Koloskova et al., 2019).

5. The authors should provide a reference for the claims around Assumption 1, such as the eigenvalue bounds. For instance, (Xiao and Boyd, "Fast linear iterations for distributed averaging") or any other material where the properties of mixing matrices are explained.

6. The convergence rate recovers that of NIDS as mentioned in Remark 3. Does it also recover the right rate in other special cases, for instance, when the algorithm is fully centralized (W=I)?

6. In figure 4, please clarify if you use the running average of the losses or the full train loss.

Typos:
Corollary 1: the expression for condition number has u instead of \mu in the denominator.
p.7, "before uniformly partitioned": should be "before being uniformly partitioned"

---

> ### Author Response · Authors · 2020-11-18
> **Response to AnonReviewer1**
>
> Dear reviewer,
>
> Thanks for the positive and constructive feedback. Here are the responses to your comments:
>
> 2. As you mentioned, $D$ is the dual variable which ensures stationary at the optimum. The purpose of introducing $H$ is to reduce the compression error. Specifically, in NIDS, the variable $Y$ should be communicated to update $D$. However, directly compressing $Y$ will cause large compression error which harms the convergence. Therefore, we maintain a state variable $H$ (updated as running average of $\hat Y$) to gradually approximate $Y$ such that the difference between $Y$ and $H$ vanishes. If the difference vanishes, according to our Assumption 2, the compression error on the difference will also decrease to 0. Note that the coarse version of Y, namely $\hat Y$, can be recovered from the compressed difference by adding $H$ back since $H$ is also maintained in the neighboring nodes. We will revise the presentation of the algorithm and make these points clear in the final revision according to your constructive suggestions.
>
> 3. We believe such relaxation is possible but it is not so straightforward based on our current analysis. It would be a good direction to pursue in the future.
>
> 4. In  our revised version, we simplify Theorem 1 a little bit and further provide Corollary 1 to clearly demonstrate the complexity bounds of LEAD and the dependency on objective function, graph connectivity as well as compression operator. We hope these will help the readers to understand our theorem easily.
> Specifically, with full-gradient, LEAD converges linearly but CHOCO-SGD can't achieve this.
> With stochastic gradient, both LEAD and Choco-SGD have the complexity $\mathcal{O}(\frac{1}{\epsilon})$. But the assumption of Choco-SGD is stronger (bounded gradient) while the result is weaker since the convergence is shown on the average model and in the ergodic sense. In contrast, the convergence of LEAD is shown agent-wise and in the non-ergodic sense. Please refer to the revised version for more details.
>
> 5. We include the references for the mixing matrix in the revision.
>
> 6. When the network is fully connected (centralized communication), we have $W=\mathbf{1} \mathbf{1}^T/n$. With full-gradient ($\sigma=0$), the complexity bound becomes $\mathcal{O} (\kappa_f \log \frac{1}{\epsilon})$, which recovers the rate of gradient descent. With stochastic gradient ($\sigma>0$), the complexity bound is $\mathcal{O}(\frac{1}{\epsilon})$ which recovers the rate of SGD.
>
> 7. In Figure 4, we plot the average loss among mini-batches within each epoch.
>
> 8. We correct the typos in the revised version.
>
> We will further carefully revise our paper based on your valuable suggestions. Thank you for your efforts to help us improve this work.

---

> > ### Comment · AnonReviewer1 · 2020-11-24
> > **Thank you for providing the rate for LEAd but please also add a rate *comparison* with the work of Koloskova et al.**
> >
> > Thank you for adding corollary 1, it is very useful. The mentioned rate of $O(\frac{1}{\epsilon})$ that is achieved by both LEAD and Choco-SGD does not have explicit constants. I believe the rate will not be the same if you include the dependency on C. Please include it in the updated version.

---

### Official Review · AnonReviewer2 · 2020-10-30
**Official Blind Review #2**

**Rating:** 7
**Confidence:** 4

**Review:**

The paper introduces a novel decentralized algorithm (LEAD) incorporated with compression that achieves linear convergence rate in strongly convex setting. The main idea is to apply and communicate the compression of an auxiliary variable instead of the primal or dual iterates.  Convergence analysis is provided for both deterministic and stochastic variants. Experiments shows the state-of-the-art performance.

The paper is well written and the results are presented clearly. I only have a few questions regarding the presentation of the algorithm.
a) It would be better to discuss in more details that the algorithm reduces to NIDS when there is no compression. In the current presentation, this point is not clearly stated in the main paper (only in appendix B).
b) The role of the auxiliary variables $Y$ and $H$ deserves a better  explanation. It is unclear to me why we apply the quantization on the difference between $Y$ and $H$, in other words, what does this difference represent? Providing more intuitions on these points will be helpful for further understanding.
c) According to the equation on the top of page 5, it seems like the quantization only introduces an additive noise in the update. Can it be viewed as an inexact variant of NIDS where the noise is bounded by some iterate dependent quantity?
d) Does the result transfer to the convex but non-strongly convex setting?
e) I am wondering whether the parameter $C$ in the contractive operator is dimension dependent, in which case would the stepsize also be dimension dependent?
f) In Figure 1 b) and Figure 2 b), some curves stop very early, it would be better to fix them.

Overall, I am very positive about the paper and hence recommend acceptance.

---

> ### Author Response · Authors · 2020-11-18
> **Response to AnonReviewer2**
>
> Dear reviewer,
>
> Thanks for the positive and constructive feedback. Here are the responses to your comments:
>
> a) 	We will state the connection with NIDS more clearly in the main paper since we have more space for the final version.
>
> b)	When there is no compression, LEAD reduces to NIDS where the variable $Y$ should be communicated to update $D$. However, directly compressing $Y$ will cause large compression error which harms the convergence. The purpose of introducing $H$ is to reduce the compression error. Specifically, we maintain $H$ (updated as running average of $\hat Y$) to gradually approximate $Y$ such that the difference between $Y$ and $H$ vanishes. If the difference vanishes, according to our Assumption 2, the compression error on the difference will also decrease to 0. Note that the coarse version of $Y$, namely $\hat Y$, can be recovered from the compressed difference by adding $H$ back since $H$ is also maintained in the neighboring nodes. Therefore, variable $H$ acts like a reference point where the compression is based on.
>
> c) 	It is true that LEAD can be viewed as an inexact variant of NIDS with the compression error being well controlled, i.e., through both difference compression and error compensation. Actually, the proposed compression and update scheme in LEAD is not restricted for the compression of NIDS but can be easily extended to other primal dual algorithms from the point of view of inexact primal-dual algorithms.
>
> d)	We believe it is not difficult to extend the convergence analysis of LEAD to the general convex setting with sublinear convergence rate. But we didn’t include this proof since we focus on achieving the linear convergence property in this paper. It will be a nice extension for our future version.
>
> e)	In general, the parameter $C$ in the contractive operator is dimension dependent. But in practice, $C$ and the step size can be dimension independent. For instance, in rank-k and top-k compression, we have $C=1-k/d$ where $d$ is the dimension of the model. However, in practice, we usually choose $k$ to be proportional to $d$, e.g., k=1% * d, so C=1-1%=0.99 which is not dimension dependent. For the $p$-norm compression used in our experiments, $C$ depends on the dimension of the vector being compressed. However, since we are able to choose the block-wise compression (e.g., to divide the whole variable into small blocks and compress each block independently), $C$ only depends on the block size we choose but not on the dimension of the model. Note that additional communication bits (i.e., one additional float number for each block) required for the blockwise compression is negligible if the block size is reasonably large.
>
>
> f)  	In Figure 1 (b) and Figure 2 (b), we plot the loss of all curves with the same number of epochs. Those curves seem to stop early because their communication bits are significantly less than the non-compressed variants such as DGD and NIDS. We can plot those curves with more epochs but that would not change the trend of those curves.
>
> We will further carefully revise our paper based on your valuable suggestions. Thank you for your efforts to help us improve this work.

---

> > ### Comment · AnonReviewer2 · 2020-11-18
> > **Re:  Response to AnonReviewer2**
> >
> > Thanks for the detailed explanation, my concerns are are clarified. It would be great to include some of these clarifications into the final version of the paper. I believe the paper deserves publication and hence maintain my score.

---

### Official Review · AnonReviewer4 · 2020-11-03
**This paper introduces a novel linearly convergent algorithm for decentralized optimization when nodes can only communicate compressed signals. Overall, I think the theoretical guarantees of the paper are strong.**

**Rating:** 7
**Confidence:** 4

**Review:**


This paper introduces a novel algorithm for decentralized optimization when nodes can only communicate a compressed signal with their neighbors. Unlike most decentralized methods with compression that are inspired by primal methods (DGD type methods), this paper introduces a new primal-dual algorithm with compression. The proposed method's main idea is borrowed from the NIDS algorithm, which converges linearly when the local loss functions are smooth and strongly convex. As the proposed LEAD method is based on primal-dual methods, it succeeds in improving the sublinear rate of primal-based methods. To the best of my knowledge, this is the first decentralized method that achieves a linear convergence rate in the setting that nodes use compressed signals.

Overall, the paper is well-written, and the authors explain the intuition behind each step of their proposed method very well. Although the main proposed method and its convergence analysis are similar to the ones in the paper that introduce the NIDS algorithm, the final theoretical result is strong. Therefore, the reviewer recommends the acceptance of this paper. A few minor comments:


It is acceptable that the authors provide a linear convergence rate for the proposed method, but it would be better to characterize the overall complexity bound for their proposed method to achieve an $\epsilon$ accurate solution. In particular, it would be great if they could study the dependency of the overall complexity (number of communication rounds) on the graph connectivity  parameter and the objective function parameters (strong convexity and smoothness constants.)


For the stochastic case, there has been a recent line of work that shows that if each local function can be written as a finite sum of a large number of functions (which is often the case in most machine learning problems), it is possible to obtain exact linear convergence rate. The current result for the stochastic setting is a bit trivial, considering the result for the deterministic setting. It would be interesting if the authors could use the tools used in the following papers to improve their results for the stochastic setting.


 1- Dual-Free Stochastic Decentralized Optimization with Variance Reduction

 2- An accelerated decentralized stochastic proximal algorithm for finite sums

 3- DSA: Decentralized Double Stochastic Averaging Gradient Algorithm

 4- Towards More Efficient Stochastic Decentralized Learning: Faster Convergence and Sparse Communication

 5- An Optimal Algorithm for Decentralized Finite Sum Optimization

---

> ### Author Response · Authors · 2020-11-18
> **Response to AnonReviewer4**
>
> Dear reviewer,
>
> Thanks for the positive and constructive feedback. Here are the responses to your comments:
>
> 1. According to your suggestion, in our revised version, we clearly demonstrate the complexity bounds of LEAD in Corollary 1. With full gradient, LEAD achieves linear convergence. The complexity bound to achieve $\epsilon$-accurate solution is $\mathcal{O}\Big ( \big ( (1+C)(\kappa_f + \kappa_g) + C\kappa_f \kappa_g \big ) \log \frac{1}{\epsilon} \Big )$ where $\kappa_f$ and $\kappa_g$ denote the conditional numbers of the objective function and the communication graph, respectively.
> When $C=0$ or $C$ is small enough, it recovers the rate of NIDS, i.e., $\mathcal{O}\Big ( (\kappa_f + \kappa_g) \log \frac{1}{\epsilon} \Big )$, and if further with centralized communication, it recovers the rate of gradient descent, i.e., $\mathcal{O} (\kappa_f \log \frac{1}{\epsilon} )$.
> With stochastic gradient, LEAD converges to the exact solution sublinearly and attains the complexity $\mathcal{O}(\frac{1}{\epsilon})$. Please refer to the revised version for more details.
>
> 2. For the stochastic case, if the local function has a finite sum structure, we believe it is highly possible to obtain exact linear convergence with techniques like variance reduction. We would like to pursue this in our future work.
>
> Thanks for the insightful suggestions and your efforts to help us improve this work.

---

### Decision · Program_Chairs · 2021-01-07
**Final Decision**

**Decision:**

Accept (Poster)

**Comment:**

The paper introduces LEAD, a decentralized optimizer with communication compression that can achieve linear convergence rate in the strongly convex setting. In terms of novelty, the authors should still add a discussion of `Magnusson et al., 2019, On Maintaining Linear Convergence of Distributed Learning and Optimization under Limited Communication`,  which is a related linear convergence result in the deterministic (full gradient) case, and relates to the analysis here which is stochastic but also exploits the deterministic case. Nevertheless, reviewers reached consensus-*with communication compression in the given time*-that the paper in its current form is well written and the results are presented clearly in both experiments and theory (which builds up on the earlier NIDS algorithm). The presentation of the algorithm can be slightly improved. We hope the authors will incorporate the remaining smaller open points such as mentioned by R1, such as making the constants in the convergence bounds explicit when comparing with other methods.